# A single m⁶A modification in U6 snRNA diversifies exon sequence at the 5' splice site

Yuma Ishigami [1], Takayuki Ohira[1], Yui Isokawa[1], Yutaka Suzuki [2] & Tsutomu Suzuki [1✉]

$N^6$-methyladenosine (m⁶A) is a modification that plays pivotal roles in RNA metabolism and function, although its functions in spliceosomal U6 snRNA remain unknown. To elucidate its role, we conduct a large-scale transcriptome analysis of a *Schizosaccharomyces pombe* strain lacking this modification and found a global change of pre-mRNA splicing. The most significantly impacted introns are enriched for adenosine at the fourth position pairing the m⁶A in U6 snRNA, and exon sequences weakly recognized by U5 snRNA. This suggests cooperative recognition of 5' splice site by U6 and U5 snRNPs, and also a role of m⁶A facilitating efficient recognition of the splice sites weakly interacting with U5 snRNA, indicating that U6 snRNA m⁶A relaxes the 5' exon constraint and allows protein sequence diversity along with explosively increasing number of introns over the course of eukaryotic evolution.

[1] Department of Chemistry and Biotechnology, Graduate School of Engineering, The University of Tokyo, 7-3-1 Hongo, Bunkyo-ku, Tokyo, Japan. [2] Department of Computational Biology and Medical Sciences, Graduate School of Frontier Sciences, The University of Tokyo, 5-1-5 Kashiwanoha, Kashiwa-shi, Chiba, Japan. ✉email: ts@chembio.t.u-tokyo.ac.jp

Post-transcriptional modification is a characteristic structural feature of RNAs. About 150 types of chemical modifications have been found in various RNAs from all domains of life[1–3]. Recent studies using deep-sequencing technologies successfully mapped several species of RNA modifications in eukaryotic mRNAs and non-coding RNAs in a transcriptome-wide manner[4–6]. These findings raise the concept of the "epitranscriptome" and highlight the importance of RNA modification as a previously unrecognized layer of regulatory gene expression.

$N^6$-methyladenosine (m$^6$A), an abundant modification in eukaryotic mRNAs and long non-coding RNAs, plays a critical role in various biological processes, including meiosis[7,8], circadian rhythm[9], sex determination[10,11], cell proliferation[12], differentiation[13], reprogramming[14], and stress responses[15]. The biogenesis and dynamics of m$^6$A have been studied extensively. The METTL3/METTL14 writer complex introduces m$^6$A in the RRACH motif (R = A or G; H = except G)[16–23]. Internal m$^6$As are abundant in the last exon of mRNAs, near the stop codon[20,21,24,25], and are decoded by several reader proteins, including YTH proteins, thereby contributing to the diverse fates of mRNAs[26–33]. m$^6$A also has the ability to destabilize double-stranded regions of structured RNA, thereby facilitating recognition by hnRNP C and hnRNP G[34,35]. It has been proposed that m$^6$A is a reversible modification that can be modified back to adenosine by erasers FTO and ALKBH5[36,37].

In addition to internal m$^6$A, $N^6$, 2′-O-dimethyladenosine (m$^6$Am) is present at the transcription start site of mRNAs in vertebrates[38,39]. Recently, our group reported a cap-specific m$^6$A writer, CAPAM/PCIF1, which catalyzes $N^6$-methylation of m$^6$Am[40]. Structural studies of CAPAM revealed the molecular basis of cap-specific m$^6$Am formation, and ribosome profiling analysis revealed that CAPAM-mediated $N^6$-methylation promotes translation of mRNAs starting from m$^6$Am.

METTL16 is an m$^6$A writer for U6 snRNA and *MAT2A* mRNA[41–43], which encodes *S*-adenosylmethionine (SAM) synthetase. METTL16 targets a specific nonamer sequence in structured RNA (UAC A GAGAA) for m$^6$A formation[41,44,45]. At high SAM concentrations, METTL16 introduces m$^6$A into six hairpin structures in the 3′ UTR of *MAT2A* mRNA, leading to degradation of the mRNA mediated by YTHDC1. At low SAM concentrations, METTL16 is bound to the hairpin structures and acts as a splicing enhancer for the 3′ terminal intron, thereby increasing production of mature *MAT2A* mRNA. Thus, METTL16 mediates a feedback mechanism that controls SAM homeostasis. This regulatory mechanism is critical for cell viability, as shown by the essentiality of *METTL16* gene in human culture cells[46] and *Mettl16* in mouse embryo[44]. *Mettl16* null mutations are embryonically lethal, and homozygotes die around the implantation stage (E3.5). Early embryos lacking *Mettl16* contain reduced amounts of *Mat2a* mRNA, highlighting the importance of Mettl16 for production of this message.

U snRNAs are essential components of the spliceosome, contributing to recognition of substrate pre-mRNAs and serving as ribozyme catalysts of two consecutive transesterifications to ligate two exons concomitant with removal of an intron[47–54]. U snRNAs contain a number of modified internal nucleosides, including pseudouridines, ribose 2′-O-methylations, and several base methylations[55–58]. 2′-O-methylation and pseudouridylation of U2 snRNA are essential for snRNP biogenesis and efficient splicing[59–65]. In U6 snRNA from mammals[56,66], *S. pombe*[57], and broad bean[67], an m$^6$A is present in the middle position of the ACAGA box that recognizes the 5′ splice site (5′SS)[56,57,66,67] (Fig. 1a and Supplementary Fig. 1). In human, the m$^6$A at position 43 in the ACAGA box is introduced by METTL16[41,42,68], whereas in *Schizosaccharomyces pombe*, the ortholog Mtl16 is responsible

for the m$^6$A modification at position 37 in the ACAGA box[41]. This m$^6$A faces the fourth position of the intron when the ACAGA box base-pairs with the 5′SS in the spliceosome B complexes, implying a functional role of m$^6$A in mRNA splicing[42]. However, because little change in global splicing is observed in null mutant mice embryos[44], and because alteration in splicing has not been studied in *METTL16*-deficient strains from other organisms, the functional significance of m$^6$A in U6 snRNA remains elusive.

To elucidate the functional role of m$^6$A in U6 snRNA, we performed RNA-seq analyses of an *S. pombe mtl16* knockout strain. We found that a subset of introns was retained in mRNAs, indicating a splicing defect caused by loss of m$^6$A in U6 snRNA. The most significantly impacted introns were enriched for adenosine at the fourth position from the 5′SS, which base-pairs with m$^6$A of U6 snRNA. In addition, the retained introns tended to have sequences that are weakly recognized by U5 snRNA at the 3′ termini of their 5′ exons, suggesting that U6 and U5 snRNPs cooperatively recognize the 5′SS, and that m$^6$A in U6 snRNA facilitates efficient splicing of a subset of pre-mRNAs.

## Results

**Physiological phenotypes of *S. pombe mtl16* knockout strains.** To elucidate the functional roles of m$^6$A in U6 snRNA (Fig. 1a and Supplementary Fig. 1), we chose *S. pombe* as a model organism because m$^6$A was undetectable in the poly(A)$^+$ RNA fraction (Supplementary Fig. 2) and the *S. pombe MAT2a* homolog *sam1* contains no hairpin structures harboring the specific nonamer sequence targeted by METTL16[69]. Consequently, U6 snRNA m$^6$A is the sole target for *S. pombe* Mtl16. First, we verified that U6 snRNA m$^6$A is introduced by *S. pombe* Mtl16. To this end, we isolated U6 snRNAs from wild-type (WT) and *mtl16* knockout (*mtl16Δ*) strains by reciprocal circulating chromatography (RCC)[70]. The isolated U6 snRNA was digested by RNase T$_1$ and subjected to capillary LC-nano-ESI-MS to analyze the RNA fragment containing m$^6$A37. m$^6$A was clearly present in U6 snRNA isolated from the WT strain (Fig. 1b and Supplementary Fig. 3a), and no fragment bearing A37 was detected, indicating that U6 snRNA was fully modified with m$^6$A37. By contrast, m$^6$A37 was completely absent in the U6 snRNA isolated from the *mtl16Δ* strain (Fig. 1b and Supplementary Fig. 3b). Next, we conducted an in vitro reconstitution of m$^6$A in U6 snRNA using recombinantly expressed *S. pombe* Mtl16 in the presence or absence of SAM. m$^6$A was efficiently introduced at position 37 of U6 snRNA in the presence of SAM (Fig. 1c), whereas m$^6$A was slightly detected even in the absence of SAM (Fig. 1c), indicating that Mtl16 contains endogenous SAM in its active site. In addition, we probed the modified RNA fragment by collision-induced dissociation (CID) analysis to confirm the presence of m$^6$A at position 37 (Supplementary Figs. 3c, d). These results demonstrated that *S. pombe* Mtl16 is an m$^6$A writer for U6 snRNA. In a previous study[43], we reconstituted m$^6$A formation in 3′ UTR hairpins of *MAT2A* mRNA using human METTL16. Similarly, in this study, we confirmed m$^6$A formation in U6 snRNA by human METTL16 (Supplementary Figs. 4a, b).

We searched for physiological phenotypes of the *mtl16Δ* strain on media containing some inducers for cellular stress. As reported previously[71], the *mtl16Δ* strain was sensitive to reagents that induce DNA damage, e.g., hydroxyurea (HU) and thiabendazole (TBZ) (Fig. 1d). In addition, we found that the *mtl16Δ* strain was sensitive to salt stress (KCl and CaCl$_2$); these sensitivities were exacerbated when the cells were cultured at 37 °C (Fig. 1d). The *mtl16Δ* strain grew more

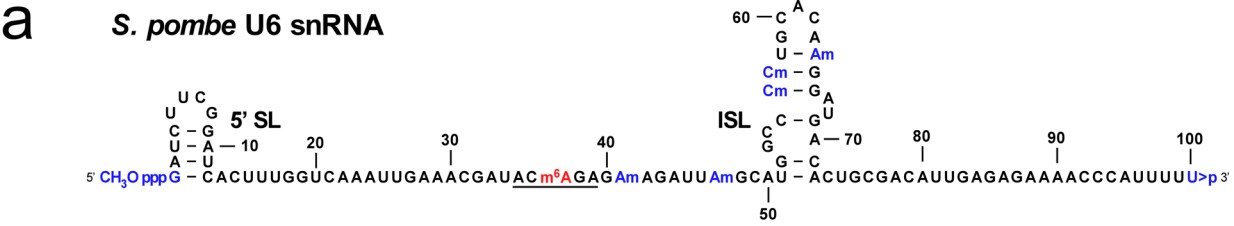

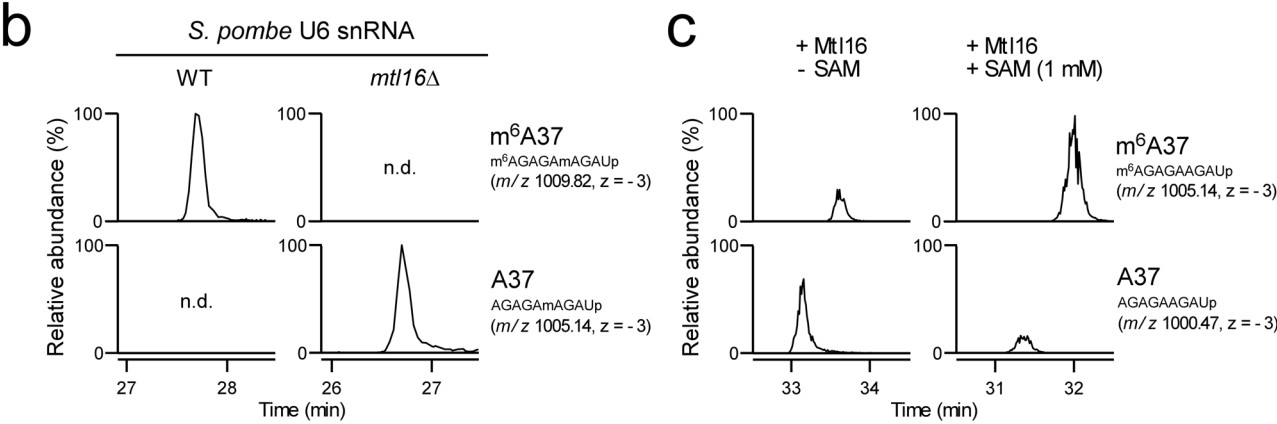

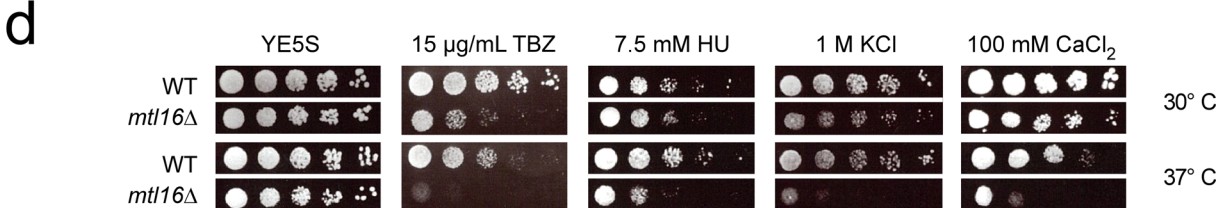

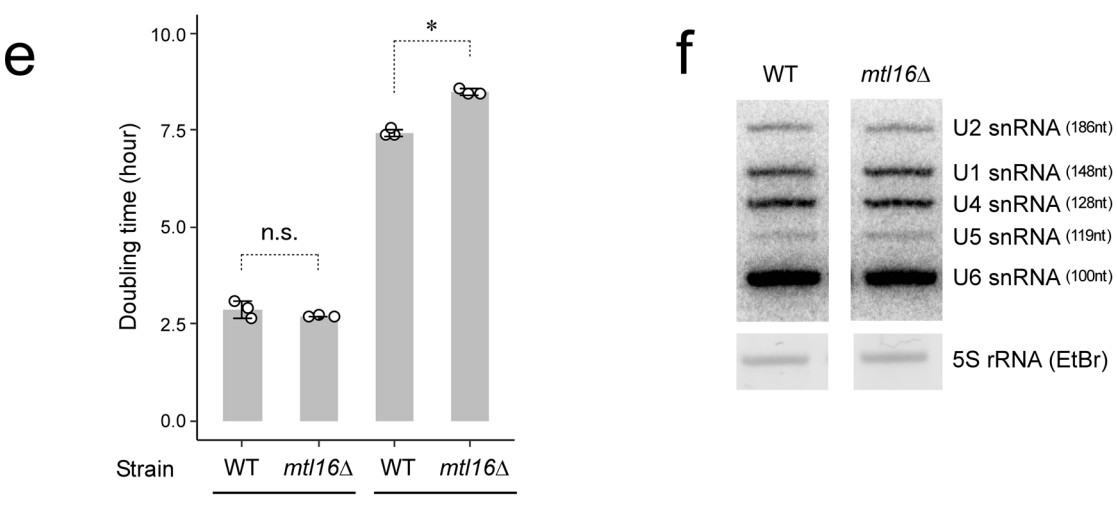

slowly than the WT strain in medium containing a non-fermentable carbon source (Fig. 1e), implying slight reduction of mitochondrial activity.

To determine whether m⁶A loss affects stability of U6 snRNA, we analyzed steady-state levels of U snRNAs by Northern blotting (Fig. 1f). We detected no significant difference in the levels of U6 and other U snRNAs between the WT and *mtl16Δ* strains,

implying that the phenotype of the *mtl16Δ* strain is due to a functional consequence of the m⁶A loss from U6 snRNA.

**Global alteration of mRNA splicing caused by m⁶A loss in U6 snRNA.** Given that m⁶A37 in U6 snRNA is present in the ACAGA box, which pairs with the 5′SS of the intron in the spliceosome, we next asked whether mRNA splicing would be

**Fig. 1 Phenotypes of cells lacking the m⁶A modification in U6 snRNA. a** Primary and secondary structures of U6 snRNA from *S. pombe* with the following post-transcriptional modifications: CH₃OpppG (γ-methyl triphosphate cap), $N^6$-methyladenosine (m⁶A), 2′O-methylations (Nm), and 2′,3′ cyclic phosphate (>p). 5′SL and ISL represent the 5′ stem loop and internal stem loop, respectively. The underlined sequence represents the ACAGA box. **b** Loss of m⁶A in U6 snRNA isolated from *SPAC27D7.08c* knockout (*mtl16Δ*) cells. Mass chromatograms of the RNase A-digested fragments of U6 snRNA containing m⁶A37 or A37 isolated from *S. pombe* WT and *mtl16Δ* cells. n.d., not detected. **c** In vitro reconstitution of m⁶A in U6 snRNA catalyzed by *S. pombe* Mtl16 in the presence or absence of SAM. Mass chromatograms of the RNase A-digested fragments of U6 snRNA containing m⁶A37 or A37. **d** Growth phenotypes of *S. pombe* WT and *mtl16Δ* strains cultured at 30 °C or 37 °C on YE5S plates supplemented with the indicated inducers of cell stress. The plates were photographed 3–4 days after inoculation. TBZ, thiabendazole; HU, hydroxyurea. **e** Doubling times of WT and *mtl16Δ* cells were measured in glucose (YE5S) and glycerol media. $OD_{600}$ was measured every hour for 12 h. The asterisk indicates a statistically significant difference in the doubling time of the two strains, as determined by two-sided Student's *t*-test. *$p$ = 7.7e-5; *n* = 3 biologically independent samples. Data are presented as mean values +/− SD. Source data are provided as a Source Data file. **f** Northern blotting of *S. pombe* U snRNAs in total RNA of WT and *mtl16Δ* cells. 5 S rRNA in total RNA was stained by EtBr as a loading control. Length of each U snRNA is indicated. This result was repeated once with a similar result. Source data are provided as a Source Data file.

affected globally by loss of m⁶A in U6 snRNA. To this end, we performed RNA-seq of poly(A)⁺ RNAs from WT and *mtl16Δ* strains and compared the splicing properties of the two samples. Based on *S. pombe* genome annotations, we first took reads that mapped to each intron and its neighboring exons and classified them into five groups (Fig. 2a): 5′ exon–intron junction reads (EIJR), intron-3′ exon junction reads (IEJR), exon–exon canonical splicing junction reads (CSR), 5′ alternative splice site selection reads (A5R), and 3′ alternative splice site selection reads (A3R). We then developed algorithms to detect alteration in splicing efficiency and transcript abundance. For each of five groups (Fig. 2a), the number of reads (coverage) is returned by function cov(x). Intron Retention Score (IRS) and Proportion of Canonical Splicing (PCS) are calculated by Eqs. (1) to (3) against each intron in each sample (Supplementary Data 1). IRS is used for measuring the difference in intron retention level and PCS is used for measuring the fraction of canonically spliced transcripts.

$$\mathrm{cov(Total)} = \frac{\mathrm{cov(EIJR)} + \mathrm{cov(IEJR)}}{2} + \mathrm{cov(A5R)} + \mathrm{cov(A3R)} + \mathrm{cov(CSR)} \tag{1}$$

$$\mathrm{IRS} = \log_2\left(\frac{\mathrm{cov(EIJR)} + \mathrm{cov(IEJR)} + 0.1}{2 \times \mathrm{cov(CSR)} + 0.1}\right) \tag{2}$$

$$\mathrm{PCS} = \frac{\mathrm{cov(CSR)}}{\mathrm{cov(Total)}} \tag{3}$$

A plot of IRS for *mtl16Δ* versus WT revealed a notable bias in the distribution; specifically, a number of introns in the *mtl16Δ* strain had higher IRS values (Fig. 2b). Therefore, in the *mtl16Δ* strain, we suggest that the pre-mRNA is dissociated from the spliceosome during the steps from the association of tri-snRNP until the first transesterification, which is at the state of B, Bact or B* complexes, leaving the intron retained. For introns with large differences in IRS, we confirmed intron retention in the *mtl16Δ* strain by semi-quantitative RT-PCR (Fig. 2c, Supplementary Fig. 5). Complementation of the *mtl16Δ* strain by ectopic expression of WT or the active-site mutant (P169A/P170A) of *mtl16*[41,44] confirmed that the alterations in splicing were dependent on the methyltransferase activity of Mtl16 (Fig. 2c).

To analyze the effect of *mtl16* knockout on gene expression, we estimated the global alteration of transcriptome (Supplementary Fig. 6a and Supplementary Data 2). Since pre-mRNA splicing is significantly affected by the *mtl16* knockout, we corrected the read counts of each transcript by multiplying PCS of introns inside the CDS, and applied them for differential expression analysis. As a result, 54 and 32 genes were up- and down-regulated, respectively, significantly over 2-fold (Supplementary Fig. 6a and Supplementary Data 2). In the up-regulated gene list, a set of genes upregulated under some stress conditions, such as

heat, osmotic, oxidation, and nutrient starvation, were enriched according to the gene ontology enrichment analysis, implying that the *mtl16Δ* strain is suffering from several cellular stresses (Supplementary Fig. 6b). We speculate that the phenotypic features of this strain might be caused by accumulation of aberrant transcripts and proteins upon *mtl16* knockout. Within the down-regulated genes in the *mtl16Δ* strain, *mrp17* (mitochondrial ribosomal protein) and *icp55* (intermediate cleaving peptidase 55) are required for mitochondrial function[72–74]. The slow-growth phenotype of the *mtl16Δ* strain under nonfermentable condition (Fig. 1e) might be explained by downregulation of these proteins. In addition, downregulation of *apl5* (AP-3 adaptor complex subunit)[75] and *pot1* (shelterin complex subunit)[76] is associated with sensitivity against hydroxyurea (HU) and/or thiabendazole (TBZ) (Fig. 1d).

**Introns bearing A at the 4th nucleotide were sensitive to m⁶A loss from U6 snRNA.** To characterize the 5′ and 3′SSs of the introns that exhibited a splicing defect, we chose a subset of introns showing large differences in IRS. To this end, we gave each intron a Z-score of a two-sample Z-test with a null hypothesis of an equal IRS, as IRS values follow normal distribution (Supplementary Fig. 7). Then, their Z-scores were sorted in descending order and split by quartiles (Fig. 2b). Sequence logos around the 5′ and 3′SSs of introns with Z-scores above the 1st quartile and below the 3rd quartile were generated by SeqLogo[77] (Fig. 2d), then the nucleotides enriched and depleted in the selected introns were shown by DiffLogo[78] (Fig. 2e). Notably, the fourth nucleotide of the 5′SS was significantly more likely to be A and less likely to be U (genomic T) in the group of introns with the highest Z-scores (Fig. 2e) than in the set of introns with the lowest Z-scores, in which T and A are equally likely at this position (Fig. 2e). By contrast, we observed no large difference in the sequence logo around the 3′SS (Fig. 2e). Given that m⁶A37 in U6 snRNA faces the fourth position of the intron during the catalytic steps of splicing[79], m⁶A loss might affect 5′SS recognition of a subset of introns bearing A at the fourth position (A4 introns).

To compare the splicing errors of introns with different nucleotides at the fourth position, we classified all introns with a sufficient read depth into four groups based on their fourth nucleotide: A4 (*n* = 2539), T4 (*n* = 1846), C4 (*n* = 514), or G4 (*n* = 157). We then plotted the IRS values for each group of introns in *mtl16Δ* versus WT strains (Fig. 3a and Supplementary Fig. 8). A large number of A4 introns exhibited a large difference in IRS implicating splicing error in the absence of m⁶A, whereas T4 introns were only slightly over the line of equal IRS, implicating little effect by m⁶A loss (Fig. 3a). Next, we calculated the difference in IRS for each group of introns upon *mtl16* knockout and generated cumulative curves (Fig. 3b). As expected, the A4 and T4 introns had the largest and smallest IRS differences, respectively. G4

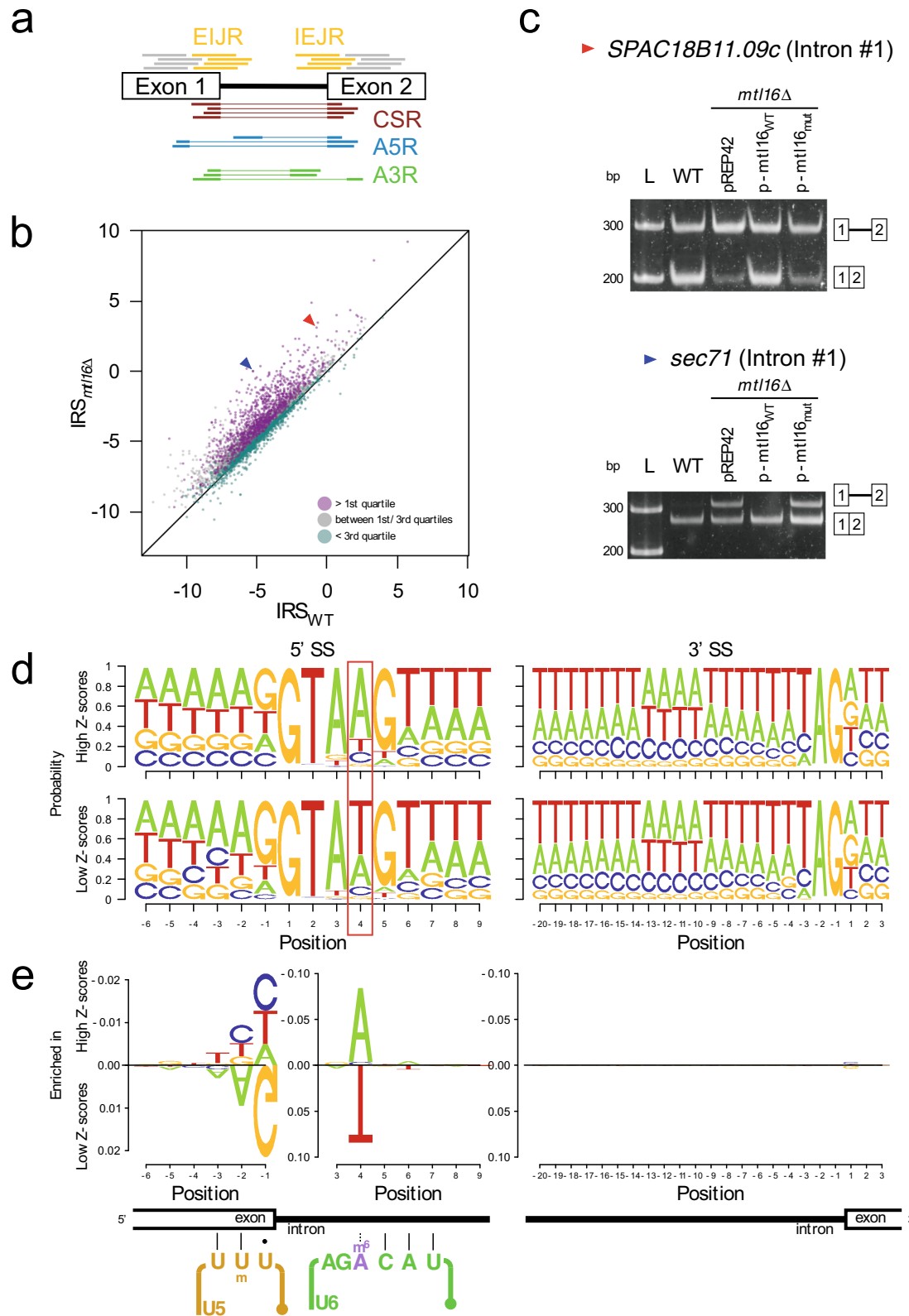

introns were less strongly affected than T4 introns, whereas the IRS differences of C4 introns were significantly higher than T4 and G4 introns, but lower than those of A4 introns. As observed for the A4 introns, the C4 introns are also spliced less efficiently than T4 and G4 introns in the absence of m6A in U6 snRNA. Thus, m6A of U6 snRNA plays a functional role in splicing of most introns

improving their efficiencies, but to a greater extent against A4 and C4 introns.

In a previous study, it was shown that an m6A protruding from a double strand stabilizes the terminal base pair through base-stacking effect enforced by the hydrophobicity of the methyl group[80]. To demonstrate this phenomenon in the spliceosome

**Fig. 2 Global alteration of mRNA splicing in *mtl16Δ* cells. a** Five classifications of RNA-seq reads mapped to each intron with its neighboring exons annotated in the *S. pombe* genome: 5′ exon–intron junction (EIJR), intron-3′ exon junction (IEJR), exon–exon canonical splicing junction (CSR), 5′ alternative splice site selection (A5R), and 3′ alternative splice site selection (A3R). **b** Scatter-plot of IRS in *mtl16Δ* versus WT. Each plot represents each annotated intron with sufficient number of reads. The plots are classified based on quartiles of the Z-score of each intron. The red and blue arrows represent plots of *SPAC18B11.09c* intron #1 and *sec71* intron #1, respectively. The black line represents an equal value of IRS. **c** Semi-quantitative RT-PCR analyses of two introns with large IRS difference. The upper and lower bands on the gel represent retained and spliced introns, respectively. Intron retention in the *mtl16Δ* strain was rescued by ectopic expression of plasmid-encoded WT mtl16 (p-mtl16$_{WT}$), but not by its active-site mutant (P169A/P170A) (p-mtl16$_{mut}$). This result was repeated once with a similar result. Source data are provided as a Source Data file. **d** Sequence logos of 5′ and 3′ splice sites with Z-scores over the top quartile (upper panels) and under the 3rd quartile (lower panels). The adenine base at the fourth position of the intron is highlighted and is associated a higher probability of intron retention in the *mtl16Δ* strain. **e** DiffLogo analysis was used to compare sequence enrichment of the 5′SS between introns with high Z-scores versus low Z-scores. The 5′SS is recognized by a part of the loop I sequence of U5 snRNA and the portion of U6 snRNA that contains m⁶A. Enriched and depleted nucleotides at the 5′SS are shown above and below the axis, respectively. The enrichment of A at position 4 against other nucleotides was significant by a p-value under $1.0 \times 10^{-98}$ by two-sided Fisher's exact test. The depletion of A, A, and G at positions −3, −2, and −1 against other nucleotides was significant by p-values under 0.01, $5.0 \times 10^{-14}$, and $5.0 \times 10^{-30}$ by two-sided Fisher's exact test, respectively. The loop I sequence of U5 snRNA and a portion of U6 snRNA pair with the 5′ exon and intron at the 5′SS, respectively. m⁶A pairs with the fourth nucleotide of introns.

context, we synthesized an in vitro transcript that mimics the interaction between the ACAGA box of U6 snRNA and the 5′SS of A4 or T4 introns, and efficiently methylated it enzymatically by recombinant human METTL16 methyltransferase domain (Fig. 3c and Supplementary Fig. 9). Then, we measured the melting temperatures of the RNA hairpins with A4 or U4 in the presence or absence of m⁶A. As a result, consistent with previous studies[80], methylation of the A–A pair to m⁶A–A increased the thermostability of the double strand, and an opposite effect was given to the A–U pair when it was methylated to m⁶A–U (Fig. 3d). This result indicates that efficient recognition of A4 introns by m⁶A is due to its thermostabilizing property in the sequence context of U6 snRNA and 5′SS.

**AAG consensus of 5′ exon suppresses sensitivity against m⁶A loss.** In all annotated introns (Fig. 2d), AAG triplet is present as a weak consensus sequence at the 3′ terminus of the 5′ exon (positions −3 to −1). This trinucleotide is complementary to loop I of U5 snRNA (Fig. 2e) and facilitates interaction between pre-mRNA and U5 snRNP in the B$^{act}$ complex[81–86]. In the Two Sample Logo (Fig. 2e), the AAG triplet was depleted in A4 introns with splicing deficiencies in the *mtl16Δ* strain, but enriched in the rest of the introns. Accordingly, we classified the A4 introns ($n = 2539$) into two groups based on 5′ exonic sequence, AAG-A4 introns ($n = 289$) and non-AAG-A4 introns ($n = 2250$); we then plotted their IRS values in *mtl16Δ* versus WT strains (Fig. 4a), which shows a notable bias only in the non-AAG-A4 introns. We further classified non-AAG-A4 introns into seven groups (BAG, ABG, AAH, BBG, BAH, ABH, and BBH-A4) (Supplementary Fig. 10a) and calculated the IRS difference between the *mtl16Δ* and WT strains for each group (Supplementary Fig. 10b). The AAG-A4 introns have the smallest ratio, whereas the BBH-A4 introns have the largest (Supplementary Fig. 10c). Thus, the farther the 5′ exon sequence is from the AAG consensus, the larger the IRS difference the intron group will bear. These results suggest that the AAG consensus of the 5′ exon alleviates the splicing deficiency of the A4 introns caused by m⁶A loss of U6 snRNA via a stable interaction with the loop I of U5 snRNA.

To confirm this observation, we carried out a mutation study of an A4 intron using a minigene construct. We chose intron 1 of the *SPAC18B11.09c* gene, whose IRS values obtained by RNA-seq are −0.733 and 3.076 in the WT and *mtl16Δ* strains, respectively (Supplementary Data 1). The IRS values of the minigene construct obtained from semi-quantitative RT-PCR were 0.058 and 3.375 in the WT and *mtl16Δ* strains, respectively (Fig. 4b and Supplementary Fig. 11). When the GU at the 5′SS was mutated to GA, no splicing took place, as expected. When we mutated A at the fourth position to U, the IRS value in the WT strain increased to the same

level as that in the *mtl16Δ* strain, indicating that the fourth nucleotide is the key cis-element for erroneous splicing in the absence of m⁶A in U6 snRNA. In addition, when the exonic ACC triplet was mutated to AAC or ACG, the splicing efficiency in both strains increased (i.e., IRS decreased). Furthermore, when the ACC was mutated to the AAG consensus sequence, splicing of this construct reached maximum efficiency even in the absence of m⁶A in U6 snRNA (Fig. 4b and Supplementary Fig. 11). This reporter experiment nicely supported our finding that m⁶A in U6 snRNA facilitates efficient splicing of A4 introns, especially when U5 snRNA is weakly associated with the 3′ terminus of the 5′ exon.

As an orthogonal approach to examine whether U6 and U5 snRNPs cooperatively recognize the 5′SS, we constructed a mutant of U5 snRNA in which UUU sequence of the loop I was replaced with AGA sequence (U5$_{AGA}$) (Fig. 4c). Then, a plasmid harboring the mutant U5$_{AGA}$ was introduced in the WT and *mtl16Δ* strains. We measured the splicing efficiencies of several A4 introns harboring non-consensus triplet sequences at the 3′ terminus of the 5′ exon (Fig. 4c). The A4 intron of *nhm1* mRNA has UCU triplet at the 5′ exonic sequence, which is weakly associated with loop I of U5 snRNA. The IRS of this intron was kept low in the WT strain, but significantly increased in the *mtl16Δ* strain as expected (Fig. 4c). When U5$_{AGA}$ was introduced, the IRS of this intron was reduced significantly, whereas WT U5 snRNA did not change its IRS (Fig. 4c), because the UCU triplet is complementary to the AGA sequence of U5$_{AGA}$. The A4 intron of *ysh1* has a UCC triplet that is recognized by the AGA in loop I of U5$_{AGA}$ with two basepairs and one mismatch. The increased IRS in the *mtl16Δ* strain was reduced significantly upon U5$_{AGA}$ expression (Fig. 4c). The A4 introns of *trk2* and *sec71* have CUG and CUC triplets at the 5′ exonic sequence, respectively. The AGA sequence of U5$_{AGA}$ weakly recognizes these triplets with single-wobble pairing at the middle of the sequence. Curiously, but as expected, the increased IRS of these A4 introns in the *mtl16Δ* strain was slightly but significantly reduced by U5$_{AGA}$ expression (Fig. 4c). As a negative control, the A4 intron of *atg1803* has a canonical AAG triplet that is stably recognized by the endogenous U5 snRNA, but not recognized by the U5$_{AGA}$ mutant (Fig. 4c). The IRS increased by U5$_{AGA}$ expression (Fig. 4c), because U5$_{AGA}$ competes with the endogenous U5 snRNA. These results clearly show that stable recognition of the 5′ exon triplet by loop I of U5 snRNA suppresses intron retention caused by m⁶A loss in U6 snRNA.

**Discussion**

Biochemical[79,81,82], genetic[87–90], and structural studies[83–86,91–94] reveal that the ACAGA box of U6 snRNA and loop I of U5 snRNA

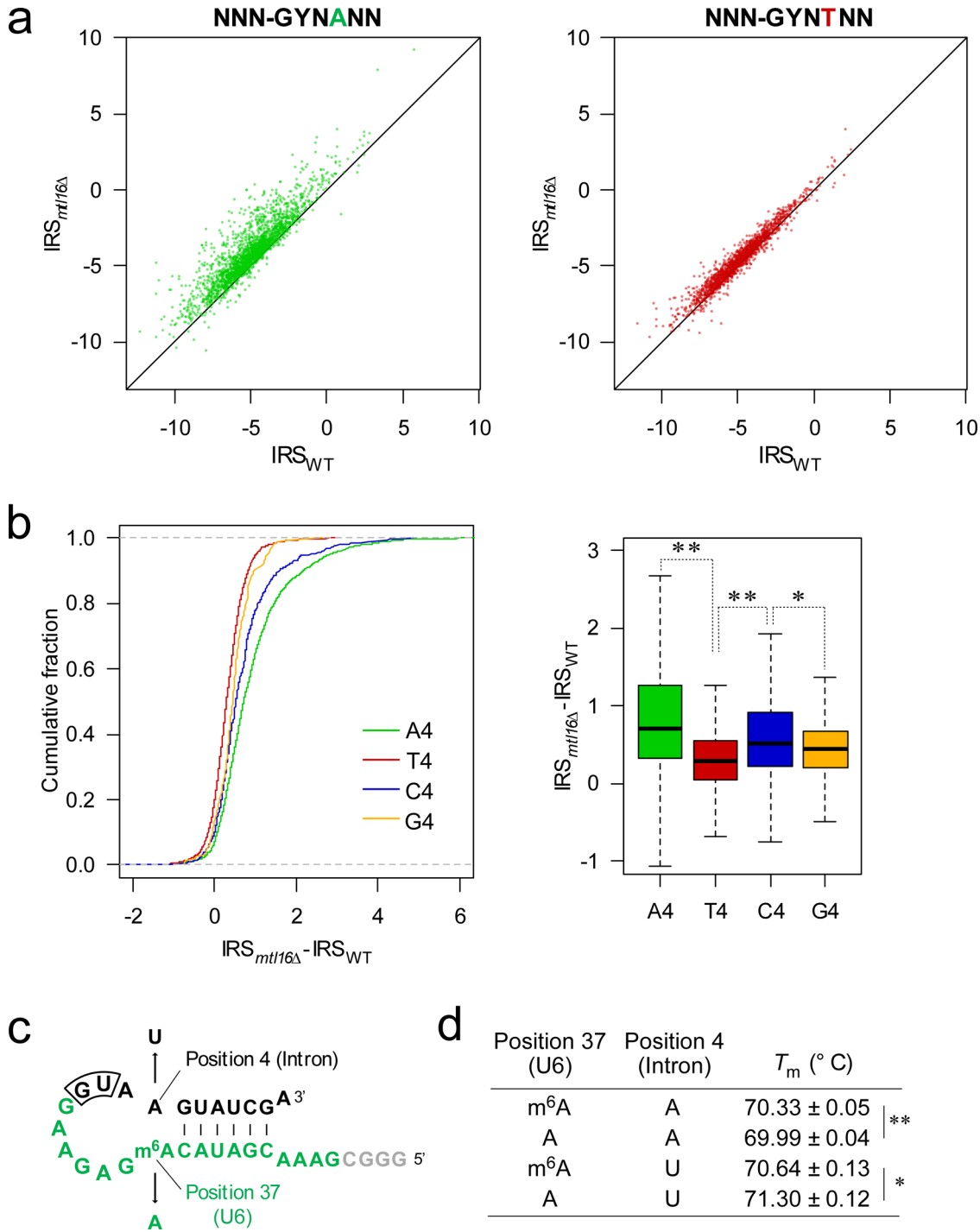

**Fig. 3 The fourth nucleotide of the intron is the cis-element for sensitivity against m6A loss. a** Scatter-plots of IRS of A4 and T4 introns in *mtl16Δ* versus WT. The black lines represent an equal value of IRS. **b** The left panel shows a cumulative plot of IRS differences for each group of classified introns in *mtl16Δ* versus WT. All introns are classified into four groups based on their fourth nucleotide. The right panel shows a box plot of IRS differences for each group of introns. The first quartile, median, and third quartile are shown, and the whiskers represent 1.5 × interquartile ranges. Sample numbers are $n = 2539$, $n = 1846$, $n = 514$, and $n = 157$ for A4, T4, C4, and G4, respectively. $*p = 6.6 \times 10^{-3}$ and $**p < 2.2e{-}16$ (two-sided Wilcoxon's rank-sum test). **c** Primary and secondary structure of the model RNA substrate and its mutation and modification introduced to the position 4 of intron and position 37 of U6 snRNA, respectively, used for measuring melting temperatures ($T_m$). Black and green characters represent the sequence derived from intron and U6 snRNA, respectively. The first dinucleotide at the 5'SS in the intron is boxed. **d** $T_m$ values of the model RNA substrates. Asterisks indicate statistically significant differences, as determined by two-sided Student's $t$-test. $**p = 7.6 \times 10^{-4}$, $*p = 3.0 \times 10^{-3}$; $n = 3$ independent experiments. Source data are provided as a Source Data file.

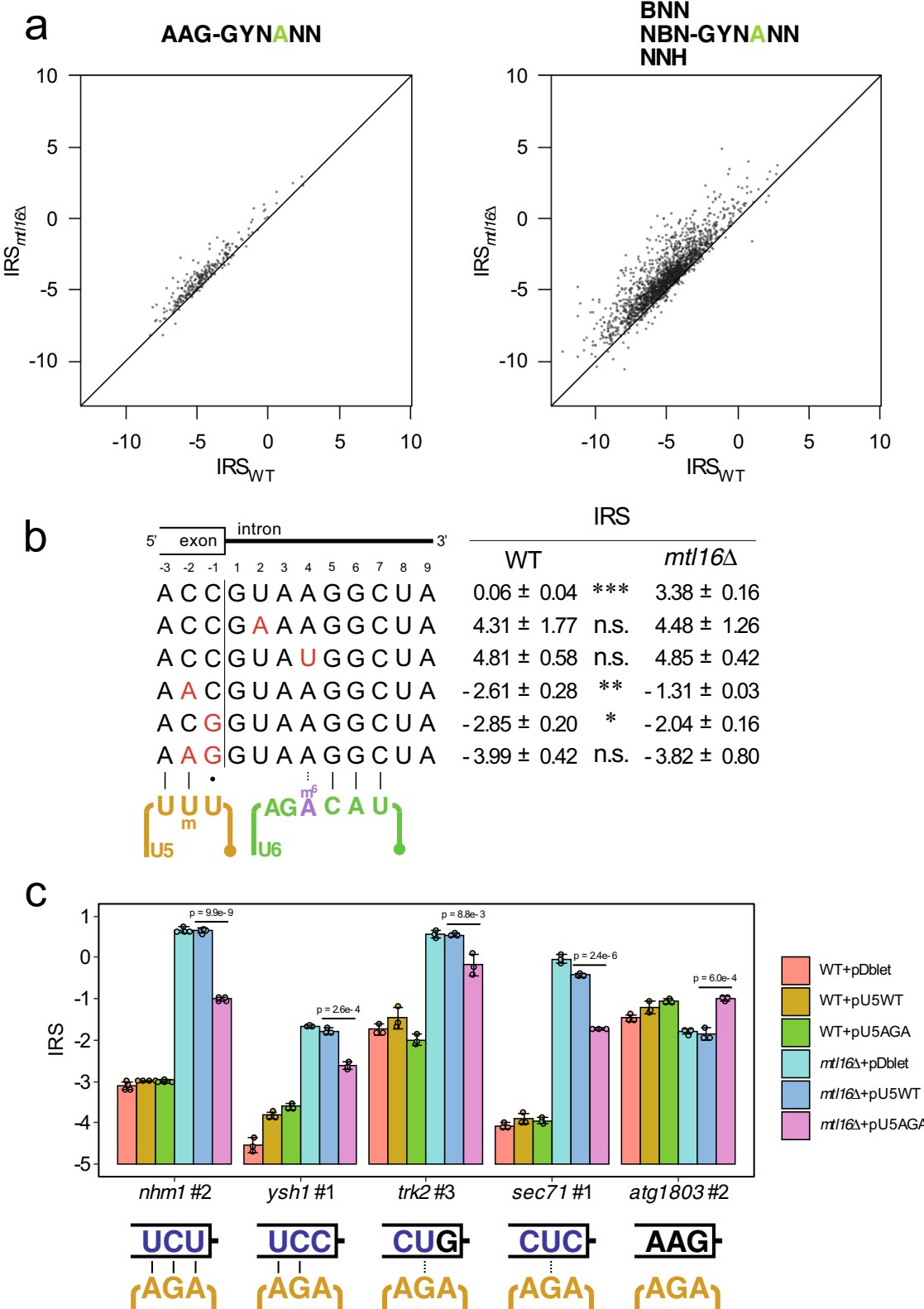

recognize 5′SS at B or B$^{act}$ complexes, and maintain this interaction until the intron lariat–spliceosome dissociation (Fig. 5a and Supplementary Fig. 12). Recognition of 5′SS by U6/U5 snRNAs contributes to definition and proofreading of pre-mRNA splicing[88,95–98], but little is known about the functional interaction between U5 snRNA–exon pairing and U6 snRNA–intron pairing. Here, we showed that in the WT *S. pombe* strain, the AAG-A4 introns are robustly recognized by U5 snRNA and U6 snRNA harboring the m$^6$A modification (Fig. 5b). Even when the AAG consensus sequence is not present in the 5′ exon, the BBH-A4 introns are stably recognized by U6 snRNA with m$^6$A modification (Fig. 5b) because m$^6$A stabilizes the interaction between U6 snRNA

**Fig. 4 Splicing defect of A4 introns was suppressed by the AAG sequence of the 5′ exon. a** Scatter-plots of IRS of AAG-A4 (left panel) introns and non-AAG-A4 introns (right panels) in *mtl16Δ* versus WT. The black line represents an equal value of IRS. **b** Mutation study with a minigene construct encoding intron #1 of *SPAC18B11.09c*. Splicing efficiency was measured by RT-PCR using minigene-specific primers, followed by non-denaturing PAGE analysis. The IRS was quantified from the signal intensity ratio of the PCR products with and without introns. Asterisks indicate statistically significant differences in IRS between the two strains, as determined by two-sided Student's *t*-test. $*p = 6.4 \times 10^{-4}$, $**p = 8.7 \times 10^{-5}$, $***p = 1.5 \times 10^{-8}$; $n = 4$ independent PCR amplifications. Source data are provided as a Source Data file. **c** IRS of several A4 introns measured by RT-qPCR for WT and the *mtl16Δ* strains transformed with plasmids. The 5′ exon triplet for each intron and its base pairing with AGA in loop I of the mutant U5$_{AGA}$ is depicted below. Asterisks indicate statistically significant differences in IRS between the two strains, as determined by two-sided Student's *t*-test. Exact p-values are written in the figure. Data represent average values of technical quadruplicates for *nhm1* #2 or technical triplicates for other introns, with s.d. Source data are provided as a Source Data file.

and the intron via m6A–A pairing. In the *mtl16Δ* strain, the AAG-A4 introns are still recognized by U5 and U6 snRNPs in B or B$^{act}$ complex, even when the m6A is not present in U6 snRNA (Fig. 5b). In this case, interaction between the 5′ exon and U5 snRNA plays a predominant role in efficient splicing. However, if the 5′ exon sequence deviates from the AAG consensus, the BBH-A4 introns are not stably recognized by U6 snRNA without the m6A modification (Fig. 5b). Collectively, these observations indicate that m6A in U6 snRNA facilitates efficient splicing of A4 introns only when U5 snRNA weakly recognizes the exon side of the 5′SS (Fig. 5b), indicating that U6 and U5 snRNPs interact cooperatively with pre-mRNA and stabilize the spliceosome during the splicing reaction. The cryo-EM structures of spliceosomes indicate that the interaction between pre-mRNA and U5/U6 snRNAs juxtaposes the 5′SS to the catalytic active site formed by the U snRNA backbones in the B complexes. Then, the Prp2 (*S. pombe* Cdc28) helicase pulls the intron to bring the branch site to the active site to form the B* complex. During these dynamic conformational changes, the 5′SS needs to be anchored to the active site, which enables the branch site 2′-OH to attack the splice site. We speculate that the stability of this anchoring interaction between pre-mRNA and U5/U6 snRNA decides the overall efficiency of the first transesterification (Fig. 5c).

As shown by the cryo-EM structures of the B and B$^{act}$ complexes (Fig. 5a and Supplementary Fig. 12a), the 3′ terminal nucleotides of the 5′ exon are recognized by the UUmU sequence of loop I in U5 snRNA, whereas the intron sequence of 5′SS is base-paired with the ACm6A sequence of the ACAGA box in U6 snRNA. The m6A of U6 snRNA faces the fourth position of the intron. In the ILS complex (Supplementary Fig. 12b), the U6 snRNA–intron pairing is slightly different from that in the B complexes. Especially, m6A37 shifts out of the pairing with U4 in the intron lariat. It is possibly due to the conformational alterations of the spliceosome after the first transesterification. Tight recognition of 5′SS by U6 snRNA might not be necessary after the lariat formation. From a chemical viewpoint, the $N^6$-methyl group of m6A has two conformations, *syn* and *anti*. Energetically, the *syn* conformation is favored over the *anti* conformation by 1.5 kcal/mol[80]. However, structural studies revealed that m6A–U pairing in the RNA duplex assumes the high-energy *anti* conformation. In support of this idea, thermodynamic measurement of RNA duplexes revealed that m6A destabilizes A–U pairs by 0.5–1.7 kcal/mol, but stabilizes A–A pairs by 0.7 kcal/mol. In addition, when m6A is present at the 5′ or 3′ overhang position, it stacks with the terminal base pair and stabilizes the RNA duplex by 0.9–1.2 kcal/mol, compared to unmodified A. Also, these thermodynamic properties were shown to be consistent with the results obtained from the sequence context of the spliceosome, by measuring the melting temperatures of modified and unmodified transcripts that mimic the base pairing between U6 snRNA and the 5′ end of an intron (Figs. 3c, d). These chemical properties of m6A neatly explain how m6A in U6 snRNA facilitates 5′SS recognition of A4 introns.

The evolutionary conservation of METTL16 implies that m6A in U6 snRNA is functionally important. METTL16 orthologs are highly conserved in metazoans, fungi, plants, and protists, implying that this modification is a fundamental component of the splicing machinery. However, m6A is not present in some organisms with small numbers of introns, including several species of budding yeasts such as *Saccharomyces cerevisiae* and *Candida albicans* and the red algae *Cyanidioschyzon merolae* (Supplementary Fig. 13). Presumably, METTL16 and m6A in U6 snRNA have been lost independently from these organisms over the course of evolution. In vertebrates, including humans, more than 200,000 introns are present in the set of all genes[99], and the consensus sequence of 5′SS is GUAAGU (A4 intron). Among fungi, *S. pombe* has over 5000 introns and *S. cerevisiae* has 296; the consensus sequences of the 5′SS are GUAAGU (A4 intron) and GUAUGU (T4 intron), respectively[69,100]. Given that m6A in U6 snRNA facilitates efficient splicing of A4 introns, *S. cerevisiae* may preferentially use T4 introns rather than A4 introns to achieve efficient splicing in the absence of this modification. *C. albicans* also has a 5′SS consensus of T4[100]. On the other hand, *C. merolae* has only 27 introns, and its 5′SS consensus sequence is GUAAGU (A4 intron)[101]. Curiously, in *C. merolae* U6 snRNA, the ACAGA box is ACUGA, and the central U base pairs with the A4 position of introns with Watson–Crick geometry[102]. This is an alternative means of stabilizing 5′SS recognition by U6 snRNA without having m6A in the ACAGA box. Based on these cases, m6A in U6 snRNA has likely contributed to expansion of 5′SS sequence variation in organisms with large numbers of introns as it facilitates efficient splicing of non-AAG-A4 introns.

METTL16 is a key regulator of SAM synthetase (MAT2A), which maintains cellular SAM homeostasis[41,43]. METTL16 introduces m6A into the hairpin structures in the 3′ UTR of *MAT2A* mRNA by sensing cellular SAM concentrations, thereby controlling mRNA stability. In addition, METTL16 acts as a splicing enhancer to recognize the same hairpins and produce functional MAT2A protein. In fact, *Mettl16* null mice die during early embryogenesis with a reduced steady-state level of *Mat2a* mRNA[44], highlighting the importance of Mettl16 function in regulation of Mat2a levels. The 3′ UTR hairpins of *MAT2A* mRNA are conserved in vertebrates, but not in other organisms[103]. In fact, *S. pombe sam1* mRNA, an ortholog of *MAT2A*, is not affected in the *mtl16Δ* strain (0.96-fold change; no significant alteration). Thus, the observed phenotypes of this strain can be explained as a function of the splicing defect caused by loss of m6A in U6 snRNA. Nematode and plant homologs of *METTL16* also have physiological functions, and strains harboring knockouts of these genes exhibit severe phenotypes. *mett-10*, a *Caenorhabditis elegans* ortholog, is required for embryonic development and germ cell differentiation[104]. *FIONA1*, an *Arabidopsis thaliana* homolog, plays an essential role in regulating the circadian clock[105]. These phenotypes might be explained by splicing alterations caused by loss of m6A from U6 snRNA.

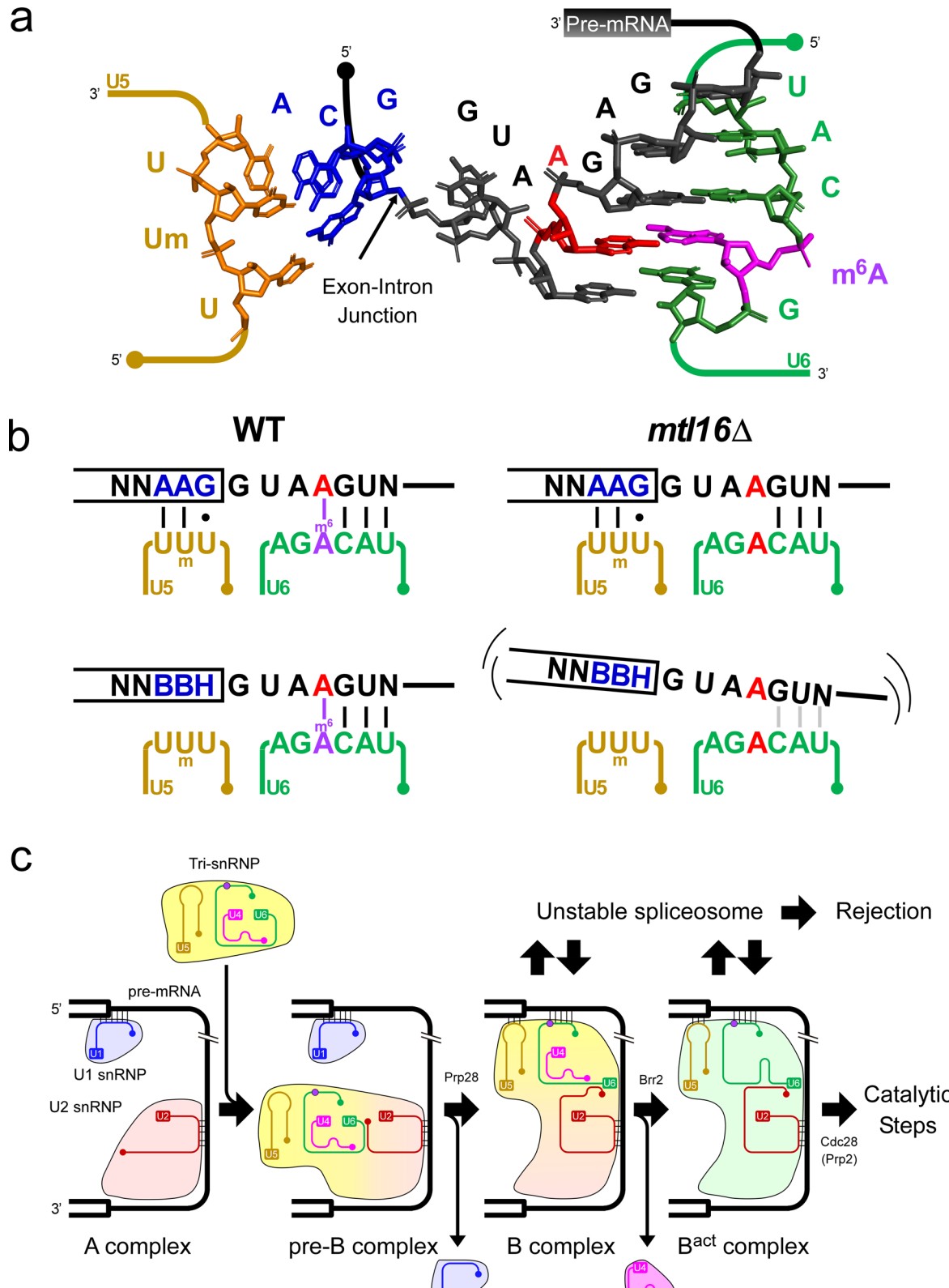

**Fig. 5 Mechanistic insight into the functional role of U6 snRNA m6A modification on pre-mRNA splicing. a** Cryo-EM structure of 5′SS in the B complex of the human spliceosome (PDB ID:5O9Z)[92], as depicted by PyMol 2.1.0 (http://www.pymol.org/2/support.html). The model consists of U5 snRNA (orange), U6 snRNA (green), 5′ exon (blue), intron (black), A4 (red), and m6A (magenta). The coordinate of the methyl group of m6A was set to its *syn* conformation. **b** Interaction between the 5′SS of A4 intron and the U5 and U6 snRNAs in WT and *mtl16Δ* strains. m6A–A pairing stabilizes the complex, especially when U5 snRNA recognizes the 5′ exon weakly. **c** Schematic depiction of spliceosomal assembly from A to Bact complex based on cryo-EM structures of *S. cerevisiae* and human spliceosomes[83–86,91,92]. The m6A base in U6 snRNA is depicted by a dark-purple circle. The blue, red, yellow, purple, and green figures represent U1 snRNP, U2 snRNP, U4/U6.U5 tri-snRNP, U4 snRNP, and Bact complex, respectively.

Because little change in global splicing is observed in early embryos (E3.5) of *Mettl16* null mice[44], the functional significance of m⁶A in U6 snRNA in mammals remains elusive. Presumably, however, maternal U6 snRNA bearing m⁶A is still present at this stage. Further studies will be necessary to determine whether m⁶A loss of U6 snRNA affects global splicing in vertebrate systems.

Given that the frequency of m⁶A in U6 snRNA is nearly 100% (Fig. 1b), this modification might be a structural component of spliceosome, relaxing the 5′ exon constraint and diversifying the protein sequence along with the number of introns increases over the course of eukaryotic evolution. However, because vertebrate METTL16 is a SAM-sensitive methyltransferase[45], and the m⁶A modifications in the 3′UTR of *MAT2A* mRNA are regulated by availability of cellular SAM[43], m⁶A in U6 snRNA might be regulated by intracellular SAM concentrations. U6 snRNA m⁶A might also be targeted by "m⁶A eraser" demethylases in higher eukaryotes. This idea raises the possibility that U6 snRNA m⁶A is not only a structural component of the spliceosome, but also a regulator of the splicing efficiency of a subset of mRNAs under SAM-depleted conditions.

## Methods

**Yeast strains and cultivation**. The ED668 (*h+ ade6-M216 ura4-D18 leu1–32*) (WT) and *SPAC27D7.08cΔ* (*mtl16Δ*) strains were obtained from the Bioneer deletion collection[74]. The deletion mutation was verified by PCR. Strains were grown at 30 °C in YE5S media [0.5% yeast extract and 3% glucose, supplemented with adenine, uracil, lysine, histidine, and leucine (225 mg/mL each)] or EMM media [EMM without dextrose (MP Biochemicals) and 2% glucose, supplemented with adenine and leucine (225 mg/mL each)]. For spot analyses, yeast cells were first grown to mid-log phase, serially diluted (1:5) from a starting sample of 10⁴ cells, and spotted onto YE5S agar plates supplemented with inducers of cellular stress: 15 μg/ml thiabendazole (TBZ), 7.5 mM hydroxyurea (HU), 1 M KCl, or 100 mM CaCl₂. The plates were photographed after 3–4 days of incubation at 30 °C or 37 °C. To measure non-fermentable growth, the yeast strains were cultured in medium containing 0.5% yeast extract, 0.1% glucose, and 3% glycerol supplemented with adenine, uracil, lysine, histidine, and leucine (225 mg/mL each). OD₆₀₀ was measured every hour for 12 h.

**RNA preparation**. Total RNA from yeast cells for snRNA isolation was prepared by hot phenol extraction[106]. Cell pellets were resuspended in extraction buffer [50 mM NaOAc (pH 5.2), 10 mM EDTA (pH 8.0), and 10% SDS], mixed with an equal volume of neutral phenol, incubated at 95 °C for 15 min, and centrifuged at 8000 × g for 30 min at 4 °C. The supernatant was collected and mixed with 1/4th volume of chloroform, followed by centrifugation at 8000 × g for 15 min at 4 °C. The supernatant was subjected to isopropanol precipitation to obtain an RNA pellet. The RNA pellet was dissolved in 10 mL buffer consisting of 20 mM HEPES-KOH (pH 7.5) and 80 mM NaCl, and then mixed with 4 mL of 2-butoxyethanol. The mixture was placed on ice for 30 min. The precipitate containing polysaccharide was removed by centrifugation at 8000 × g for 30 min at 4 °C. The supernatant was subjected to ethanol precipitation to obtain total RNA.

**Nucleoside analysis of poly(A)⁺ RNA**. Total RNA extracted from exponentially growing *S. pombe* WT cells underwent four rounds of enrichment of poly(A)⁺ RNA using Dynabeads oligo [dT]₂₅ (Invitrogen). Eluate RNAs from each round of enrichment (1 μg each) were digested into nucleosides with 0.07 units of nuclease P₁ (Fujifilm Wako Pure Chemical Corporation), 0.127 units of phosphodiesterase I (Worthington Biochemical Corporation), and 0.08 units of bacterial alkaline phosphatase (BAP from *E. coli* C75, Takara). The digests were subjected to LC/MS analysis using a Q Exactive Hybrid Quadrupole-Orbitrap Mass Spectrometer (Thermo Fisher Scientific) equipped with a Dionex UltiMate 3000 LC System (Thermo Fisher Scientific) and an InertSustain C18 column (5 μm, 2.1 × 250 mm, GL Science). The mass chromatogram peak area of adenosine (A, *m/z* 268.10), $N^6$-methyladenosine (m⁶A, *m/z* 282.12), $N^6$-dimethyladenosine (m⁶,⁶A, *m/z* 296.14), and 2′-O-methyladenosine (Am, *m/z* 282.12) was measured. Then, the peak area of m⁶A, m⁶,⁶A, and Am was divided by the peak area of A, and these ratios at each round of enrichment were normalized by the ratio from the first eluate and then plotted for each modification. Data were analyzed on Xcalibur 4.1 and Excel 2016, and visualized by R 4.0.3 and Canvas 15.

**Isolation of U6 snRNA and mass spectrometry**. Isolation of yeast U6 snRNA was performed by RCC[70] using a DNA probe (Supplementary Data 3). Mass spectrometric analysis of RNase T₁- or A-digested fragments of the isolated U6 snRNA was performed by capillary LC-nano-ESI-MS[107]. Mongo Oligo Mass

Calculator v2.08 (https://mods.rna.albany.edu/masspec/Mongo-Oligo) was used for assignment of the product ions in CID spectra[108].

**Preparation of transcripts for in vitro reconstitution**. DNA templates for in vitro transcription were prepared using a series of DNA oligos (described in Supplementary Data 3). U6 snRNAs were transcribed by T7 RNA polymerase in vitro essentially as described[109]. Briefly, DNA template was mixed with in-house made T7 RNA polymerase, NTPs, and a transcription buffer consisting of 40 mM Tris-HCl (pH 7.5), 24 mM MgCl₂, 5 mM DTT, 2.5 mM Spermidine, and 0.01% Triton X-100, and was incubated at 37 °C overnight. The reaction was stopped by addition of an equal volume of phenol/chloroform/isoamyl alcohol (25:24:1, v/v/v). The supernatant was desalted by PD-10. Desalted RNA was collected by ethanol precipitation. The transcript was resolved by 10% denaturing PAGE and purified by excision from the gel.

**Expression and purification of recombinant proteins**. To obtain recombinant *S. pombe* Mtl16, the *mtl16* cDNA was amplified by RT-PCR of total RNA from WT *S. pombe* cells using the primers listed in Supplementary Data 3, which were designed based on genomic sequence retrieved from PomBase[110]. Because the original initiation codon (genomic position I:4,522,948-4,522,950) of *mtl16* ORF in the database was misannotated inside an unannotated intron (genomic position I:4,522,922–4,522,964), we constructed a recombinant Mtl16 starting from an AUG codon (genomic position I:4,523,030–4,523,032) upstream of the unannotated intron. The PCR product was cloned into the *NheI* and *XhoI* sites of pET-28a(+) (Novagen) to yield pET-Spmtl16, which produces N-terminal hexahistidine-tagged Mtl16 (His₆-Mtl16). The *E. coli* Rosetta2 (DE3) strain was transformed with pET-Spmtl16 and cultured in LB medium containing 50 μg/mL kanamycin and 30 μg/mL chloramphenicol. When the bacteria reached OD₆₀₀ of 0.5, protein expression was induced by addition of 10 μM IPTG and the cells were grown for an additional 3 h at 37 °C. Cells were harvested and suspended in buffer A [50 mM HEPES-KOH (pH 7.5), 150 mM NaCl, and 7 mM β-mercaptoethanol] supplied with 0.5 mM PMSF, followed by sonication on ice. Cell lysates were cleared by ultracentrifugation at 100,000 × g for 60 min. The supernatant was loaded onto a nickel-charged HiTrap chelating column (GE Healthcare) and unbound proteins were washed out with buffer A. The recombinant proteins were eluted with a 50-mL linear gradient comprising 0–500 mM imidazole in buffer A. A fraction containing the recombinant protein was dialyzed overnight against dialysis buffer [50 mM HEPES-KOH (pH 7.5), 150 mM NaCl, 10% glycerol, and 7 mM β-mercaptoethanol]. The concentration of the purified protein was determined in a Bradford assay (Bio-Rad, Hercules, CA, USA) using bovine serum albumin as a standard. To obtain recombinant human METTL16, the *METTL16* cDNA was amplified using the primers listed in Supplementary Data 3, which were designed based on human genomic sequence retrieved from Ensembl[111] and cloned into pET-28a(+) to yield pET-hMETTL16, which produces C-terminal hexahistidine-tagged METTL16 (METTL16-His₆). Expression and purification were carried out as described for *S. pombe* Mtl16. To obtain recombinant human METTL16 MTD, cDNA coding the methyltransferase domain of METTL16 was amplified using the primers listed in Supplementary Data 3 and cloned into pE-SUMO-TEV by SLiCE to yield pE-SUMO-hMETTL16MTD. Expression was carried out as described for *S. pombe* Mtl16. Following expression, cells were harvested and suspended in buffer A [50 mM HEPES-KOH (pH 7.5), 150 mM NaCl, and 7 mM β-mercaptoethanol] supplied with 0.5 mM PMSF, followed by sonication on ice. Cell lysates were cleared by ultracentrifugation at 100,000 × g for 60 min. The supernatant was loaded onto a nickel-charged HiTrap chelating column (GE Healthcare) and unbound proteins were washed out with buffer A. The recombinant proteins were eluted with a 50-mL linear gradient comprising 0–500 mM imidazole in buffer A. A fraction containing the recombinant protein was dialyzed overnight against dialysis buffer [50 mM HEPES-KOH (pH 7.5), 150 mM NaCl, 10% glycerol, and 7 mM β-mercaptoethanol] containing SUMO protease Ulp1. Dialyzed protein was applied to Ni Sepharose 6 Fast Flow (GE Healthcare) and the flow-through was collected. The cleaved protein was further purified on a HiScreen Capto SP ImpRes (GE Healthcare) column. The concentration of the purified protein was determined in a Pierce BCA Protein Assay (Thermo Fisher Scientific) using bovine serum albumin as a standard.

**In vitro reconstitution using recombinant proteins**. In vitro methylation of U6 snRNA was performed at 37 °C for 1 h in 50 μL of reaction mixture consisting of 10 mM HEPES-KOH (pH 7.5), 100 mM NaCl, 1 mM DTT, 2 μM U6 snRNA transcript, 1 μM His₆-Mtl16, and 1 mM SAM. The reaction was stopped by addition of an equal volume of phenol/chloroform/isoamyl alcohol (25:24:1, v/v/v). RNA was collected by ethanol precipitation, digested with RNase T₁, and subjected to RNA-MS analysis as described above.

**Northern blotting**. Total RNA (1 μg) from yeast cells was resolved by 10% denaturing PAGE and either stained with SYBR Gold (Invitrogen) or blotted onto a nylon membrane (Amersham Hybond N+, GE Healthcare) in 1 × TBE using a Transblot Turbo (Bio-Rad). Fluorescence was visualized on an FLA-7000 imaging analyzer (Fujifilm). The membrane was air-dried and irradiated twice with UV light (254 nm, 120 mJ/cm²; CL-1000, UVP) to cross-link the blotted RNA. Hybridization was performed using PerfectHyb (TOYOBO) at 52 °C with 4 pmol of 5′-³²P-labeled oligonucleotides specific to each U snRNA (Supplementary

Data 3). The membrane was washed four times with $2 \times$ SSC [300 mM NaCl, 30 mM trisodium citrate (pH 7.0)] containing 0.1% SDS, dried, and exposed on an imaging plate (BAS-MS2040, Fujifilm). Radioactivity was visualized on the FLA-7000. Uncropped scans are available in the Source Data file.

**RNA-seq sample preparation**. Yeast were harvested, frozen in liquid nitrogen, and homogenized in TriPure Isolation Reagent (Roche Life Science) using SK-100 (Funakoshi). Total RNA was extracted with chloroform, recovered from the aqueous phase by isopropanol precipitation, ethanol precipitated, and dissolved in water. The total RNAs underwent two rounds of enrichment of poly(A)$^+$ RNA using Dynabeads oligo [dT]$_{25}$ (Invitrogen) and were sequenced on an Illumina NovaSeq sequencer (150-bp, paired-end).

**RNA-seq read alignment**. Raw Illumina sequencing reads were trimmed using fastp software[112] to remove adaptor sequences and low-quality bases. The processed reads were first aligned to *S. pombe* rRNA and tRNA sequences, then the unmapped reads were aligned to *S. pombe* genome assembly ASM294v2.34 using STAR 2.7.3a[113]. The following parameters were used–twopassMode Basic–outSJfilterOverhangMin 30 8 8 8–alignIntronMax 2000–outFilterMultimapNmax 30–outFilterScoreMinOverLread 0.4–outFilterMatchNminOverLread 0.44–outFilterScoreMin 10. The statistics of trimming and mapping is listed in Supplementary Data 4. Read alignment files (.bam) were processed with custom Python scripts[114] using the libraries Biopython[115] and Pysam[116], and the plots were written by R 4.0.3. For each segment comprising a pair of consecutive annotated exons (exon1 and exon2), reads that mapped to the annotated SS junction (CSR), exon 1–intron junction (EIJR), intron–exon 2 junction (IEJR), any alternative 5′SS to exon 2 (A5R), and exon 1 to any alternative 3′SS (A3R), were counted.

**Calculation of erroneous splicing**. For each annotated segment described above, we obtained IRS by Eq. (2) to assess retention level for each annotated intron. PCS was calculated by Eq. (3) to assess the proportion of canonically spliced mRNA within a mixture of mature, immature, and aberrant mRNA. To compare the difference in IRS between WT and $mtl16\Delta$ strains, as the IRS values mostly follow normal distribution (Supplementary Fig. 7), $Z$-scores with a null hypothesis of zero difference in IRS for each intron were obtained by Eq. (4), where $\mu$ and $\sigma$ represent the mean and standard deviation, respectively, of the IRS values among the quadruplicates of WT and $mtl16\Delta$ strains.

$$Z = \frac{\mu_{mtl16\Delta} - \mu_{WT}}{\sqrt{\sigma_{mtl16\Delta}^2/4 + \sigma_{WT}^2/4}} \quad (4)$$

All annotated GY-AG introns with a sufficient arbitrary number of reads in all four replicates [cov(Total) > 10] and an average CSR over 10 in either strain were applied for this calculation so that the number of introns picked up is maximized and the variance in read counts among the replicates is minimized. This process picked up 5056 GY-AG introns with sufficient read counts out of 5364 annotated introns. The 5′ and 3′SS sequences of the introns with splicing deficiencies were characterized by choosing a subset of introns with large and small IRS differences. Upon sorting the introns by $Z$-scores in a descending order, those over the 1$^{st}$ quartile were treated as introns with large IRS differences, and those under the 3$^{rd}$ quartile were treated as introns with small IRS differences. Then, sequence logos around the 5′ and 3′ SSs of introns with large or small IRS differences with sufficient read depths were generated by SeqLogo[77]. Then, two logos were compared by DiffLogo[78]. The raw count and result of the calculation is provided in Supplementary Data 1.

**Differential expression analysis**. Raw counts for each annotated transcript were determined using the htseq-count command of HTSeq version 0.6.1[117]. The splicing-adjusted read counts of each annotated gene in each sample were calculated by Eq. (5), where $i$ is the intron number and PCS$_k$ is the PCS value of intron number $k$. The product of each ratio of canonically spliced reads for each intron inside the CDS of the gene was multiplied against the overall read count that falls inside each genomic locus of the transcript.

$$\text{Adjusted counts} = \text{Raw counts} \times \prod_{k=1}^{i} (\text{PCS}_k) \quad (5)$$

Differential expression of adjusted counts was evaluated by edgeR version 3.12[118,119], applying a statistical cutoff of FDR < 0.05 and fold-change > 2 to obtain differentially expressed genes (Supplementary Data 2). Enriched features of gene lists were searched by the AnGeLi[120] tool with protein-coding genes as a background (Supplementary Data 5).

**Rescue experiment and intron-specific semi-quantitative RT-PCR**. The *S. pombe mtl16* gene, including untranslated regions, was PCR-amplified from the genome and cloned into pREP42-mcs+ (Addgene plasmid 52691) digested by NdeI and BamHI using the primers listed in Supplementary Data 3 and the SLiCE method[121] to yield the WT plasmid p-mtl16$_{WT}$. To construct an active-site mutant, both P169 and P170 were mutated to Ala residues by site-directed mutagenesis using the primers listed in Supplementary Data 3 to yield p-mtl16$_{mut}$. *S. pombe mtl16$\Delta$* strain was transformed with the WT or mutant plasmid by the lithium acetate method[122], cultivated in EMM medium devoid of thiamine until the cell

density reached OD$_{600}$ = 1.0, and harvested. Total RNA was prepared using the TriPure Isolation Reagent (Roche Life Science). One microgram of total RNA from WT, KO, and transformed strains were treated with RQ1 RNase-free DNase (Promega) and subjected to reverse transcription with random hexamer and oligo dT primers using the Transcriptor First Strand cDNA Synthesis Kit (Roche), followed by PCR amplification with intron-specific primers (Supplementary Data 3) under the following conditions: 30 cycles at 95 °C for 30 s, 53 °C for 45 s, and 68 °C for 30 s. PCR products were resolved on 8% non-denaturing PAGE, stained with EtBr, and visualized on FLA-7000.

**Minigene analysis**. The flanking sequence of the first intron of *SPAC18B11.09c* was amplified using primers bearing NdeI and BamHI sites (Supplementary Data 3). The PCR product was cloned into the corresponding sites of pREP42-mcs + to yield the WT minigene plasmid pREP-AC18B11.09c. Mutations were introduced by site-directed mutagenesis using the primers listed in Supplementary Data 3. *S. pombe* WT and *mtl16$\Delta$* strains were transformed with the WT or mutated plasmid using the lithium acetate method, cultivated in EMM medium devoid of thiamine, until the cell density reached OD$_{600}$ = 1.0, and harvested. Total RNA was prepared using the TriPure Isolation Reagent. One microgram of total RNA was treated with RQ1 RNase-free DNase and then subjected to reverse transcription with nmt1 promoter-specific RT primer (Supplementary Data 3) using the Transcriptor First Strand cDNA Synthesis Kit, followed by PCR amplification with minigene-specific primers (Supplementary Data 3) under the following conditions: 16–23 cycles at 95 °C for 30 s, 53 °C for 45 s, and 68 °C for 30 s. The number of cycles was adjusted so that the signals of the PCR products were not saturated. The PCR products were resolved on 8% non-denaturing PAGE, stained with EtBr, and visualized on FLA-7000. Signals corresponding to spliced or retained introns were quantified using the Multi Gauge V3.0 software (Fujifilm). Signal intensities were converted to levels of PCR product using the signals of Loading Quick 100-bp DNA Ladder (Toyobo) as a standard. IRS is the binary logarithm of the value obtained by dividing the abundance of the retained isoform by the abundance of the spliced isoform.

**Intron retention-specific RT-qPCR**. The *LEU2+* marker of pTN69 was replaced by ura4+ using the primers listed in Supplementary Data 3 and the SLiCE method[121] to yield pDblet. The *S. pombe snu5* gene and its flanking region was PCR-amplified from the genome and cloned into pDblet digested by HindIII using the primers listed in Supplementary Data 3 and the SLiCE method[121]. Then, site-directed mutagenesis using the primers listed in Supplementary Data 3 was used to introduce an additional 15-nt stem-loop in the 3′ end of U5 snRNA to yield the plasmid pU5$_{WT}$. Site-directed mutagenesis using the primers listed in Supplementary Data 3 was utilized to introduce a TTT to AGA mutation in the snu5 gene of pU5$_{WT}$ to yield pU5$_{AGA}$. Total RNA (2 μg) was extracted from WT and the *mtl16$\Delta$* strains transformed with pDblet, pU5$_{WT}$ or pU5$_{AGA}$ grown to mid-log phase in EMM media without uracil. cDNA was synthesized by Superscript III (Thermo Fisher Scientific) using random N$_6$ primers, according to the manufacturer's instructions. PCR was performed in a 10 μL mixture containing a 0.2-μL aliquot of the cDNA solution, 0.2 μM of each PCR primers, and 1 × KAPA SYBR FAST Master Mix optimized for LightCycler480 (Kapa biosystems). The thermal cycling conditions included 45 cycles of 95 °C for 10 s, 58 °C for 20 s, and 72 °C for 1 s. Amplification of DNA was monitored by a LightCycler480 (Roche), according to the manufacturer's instructions. To write a standard curve of Cp (crossing point) value against the amount of the template, depending on each set of primers, a dilution series of the *S. pombe* WT genomic DNA was amplified by the primers. To measure the IRS value of a single intron, primers spanning intron and exon were used to obtain the Cp value of the retained isoform, and primers constructed inside an adjacent single exon was used to obtain the Cp value of the total of both retained and spliced isoforms. Then, the abundance of cDNA derived from each isoform was calculated based on the standard curve. IRS is the binary logarithm of the value obtained by dividing the abundance of the retained isoform by the abundance of the spliced isoform.

**Melting temperature observation of the transcript**. In vitro transcripts U6A4, U6U4, and U6guide were synthesized from templates U6modelA4_T7R, U6modelU4_T7R, and U6modelGuide_T7R, respectively (Supplementary Data 3) as described. U6A4 and U6U4 were annealed with U6guide by incubating at 95 °C for 5 min and ramped at −0.1 °C/sec to 10 °C in a mixture consisting of 100 mM NaCl, 20 mM HEPES-KOH (pH 7.5), 75 μM U6A4 or U6U4 RNA, and 75 μM U6guide RNA. The U6guide RNA was designed to form a secondary structure with U6A4 or U6U4 RNAs that emulates human U6 snRNA secondary structure recognized by METTL16[68]. The annealed mixture was applied for methylation by incubating at 37 °C overnight in a mixture consisting of 200 mM NaCl, 50 mM HEPES-KOH (pH 7.5), 1 mM DTT, 2.5 mM SAM, 0.2 units/uL SUPERase-In RNase inhibitor (Thermo Fisher Scientific), 30 μM human recombinant METTL16 MTD, and 30 μM RNA. The reaction was stopped by mixing with an equal volume of phenol/chloroform/isoamyl alcohol (25:24:1, v/v/v). The RNA was collected by ethanol precipitation, then U6A4 RNA or U6U4 RNA was resolved by 10% denaturing PAGE and excised from the gel piece. The methylation efficiency was verified by LC/MS as described. The methylated or unmodified transcripts were dissolved at a

final concentration of 3 μM in degassed buffer consisting of 20 mM sodium phosphate (pH 7.0) and 1 M NaCl, followed by incubation at 95 °C for 5 min and ramped at −0.1 °C/sec to 10 °C for annealing. A UV–Vis spectrophotometer (V-630, JASCO) equipped with an 8 multiquartz microcell array (path length: 10 mm, JASCO) was used to monitor the hyperchromicity by monitoring absorbance at 260 nm. The temperature gradient was as follows: 25 °C for 30 sec, ramped at 5 °C/min to 40 °C and held for 5 min, and after that ramped at 0.5 °C/min to 105 °C. The measurement was performed in triplicate for each sample.

**Reporting summary**. Further information on research design is available in the Nature Research Reporting Summary linked to this article.

## Data availability

The sequence data from this study have been submitted to the DDBJ Sequence Read Archive under accession number DRA009909. Structural data was retrieved from PDB ID: 5O9Z [https://doi.org/10.2210/pdb5o9z/pdb], 5GM6 [https://doi.org/10.2210/pdb5gm6/pdb], and 3JB9 [https://doi.org/10.2210/pdb3jb9/pdb]. Source data are provided with this paper.

## Code availability

Python scripts written for RNA-seq data analysis are available from a public GitHub repository https://github.com/YumaIshigami/irscalc[114].

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

## Acknowledgements

We thank members of Tom Suzuki's laboratory, especially Yuriko Sakaguchi, for RNA-MS analysis, Kenjyo Miyauchi for RNA isolation with RCC, and Shunpei Okada for NGS data analysis. We thank Terumi Horiuchi and Kiyomi Imamura of Yutaka Suzuki laboratory for sample preparation, data generation, and data registration of NGS. Some computations were performed on the NIG supercomputer at the ROIS National Institute of Genetics. This work was supported by Grants-in-Aid for Scientific Research on Priority Areas from the Ministry of Education, Culture, Sports, Science, and Technology of Japan (MEXT), Japan Society for the Promotion of Science (JSPS) [20292782 and 18H05272 to T.S., 16H06279 (PAGS) to Y.S., and 18J13582 to Y.Ish.], and Exploratory Research for Advanced Technology (ERATO, JPMJER2002 to T.S.) from Japan Science and Technology Agency (JST).

## Author contributions

Y.Ish. performed biochemical, genetic, and bioinformatic works assisted by T.O. Y.Iso performed the poly(A)$^+$ purification-nucleoside analysis experiment. Y.Ish. and T.S. wrote this paper. Y.S. performed NGS. All authors discussed the results and revised this paper. T.S. designed and supervised all work.

## Competing interests

The authors declare no competing interests.
