## [Peer Review File · Nature Communications]

Reviewers' comments:

Reviewer #1 (Remarks to the Author):

In this manuscript, Ishigami and co-workers examine the role of an m6A modification on the U6 snRNA and its impact on splicing efficiency. The authors begin with a demonstration that a *mtl16* knockout strain lacks the m6A mark on the U6 snRNA by Mass Spectrometry. They further demonstrate that cells lacking *mtl16* are more susceptible to cellular stress but that it is not due to a change in amount of snRNPs present in the cell. To better understand the role of the m6A modification of U6 on splicing, they perform RNAseq to identify splicing events that are affected in the absence of *mtl16*. Authors' analysis of these data suggest to them a difference in the behavior of introns containing a T at position 4 of the intron relative to those containing an A. Authors then build a model correlating sequences at the 3' end of the upstream exon and the nucleotide identity at position 4 of the intron and how methylation of U6 would impact these different events.

Although I find authors' hypotheses about the role of this m6A in splicing to be quite intriguing, I cannot recommend publication of this work in its current state because of the absence of any statistical rigor in the analysis of the data presented here, and the errors that likely derive from this absence. It is well understood today that the 'noise' that is present in a quantitative sequencing experiment is not equally present for all measured individuals in an experiment, but rather is correlated with the level of detection of the individual: highly sampled events are subject to lower variance than are poorly sampled events. This phenomenon is well described in Anders and Huber, *Genome Biology* 2010. In the current work, authors use a 'home-made' method for attempting to assess meaningful changes, wherein they assess (among a few other criteria) whether " $PSE\ KO - PSE\ WT > 2 \times SD$ " (page 17, I also note that authors have an error in their formula for SD on this page in that they are missing the $\wedge 2$ for both of the individual variance values). There are at least two important problems with this approach. First, for an experiment such as this where there are only two replicates, the use of 2 SD units is massively below the requirement for using a standard t-test to assess significance. Very briefly, this is the reason that analysis tools such as dEseq use estimates of global (expression-dependent) variance in their testing rather than measured variance for individuals in their approach. Second, and equally importantly, the approach authors take incorrectly assumes that variance (SD) will be consistent across the spectrum of samples. But because variance is correlated with overall sampling level, lowly expressed samples will naturally be subject to greater variance, making it more difficult to 'significantly' detect a defect in splicing according to these criteria. In the plot below, I have taken the raw data (author provided) and calculated the coefficient of variation (SD/Mean) of the EIJ metric (for the CON1 and CON2 measurements) for each (non-zero) event in the 'CON' dataset and plotted these values as a function of the 'EIJCON1' value. (see attached file)

As expected, there is significantly higher variance associated with the lower sampled individuals (this is particularly apparent as count numbers decrease from 100 \rightarrow 1, there is also a surprising anomaly in the data above \sim 100 which suggests an artifact in normalization and/or compression). The problem with the approach that authors have used here is that it is much more difficult according to their formula to detect a lowly sampled event (those with higher overall SD) than for a highly expressed event (with lower overall SD). This problem is particularly important given that authors' data also show that global expression values for introns containing a T at position 4 are higher than are global expression values for introns containing an A at position 4, the combined result being that authors' statistical approach makes it more likely to identify T4 introns than A4 introns. This statistical problem makes it impossible to determine whether the result that authors have described in their work is actually a function of the loss of m6A, or simply reflects the increased ability of the T4 introns to pass authors' statistical filters: but that both T4 and A4 introns are equally impacted by the absence of m6A, contrary to their presented conclusion.

Reviewer #2 (Remarks to the Author):

One widespread RNA modification is N6-methylation of the nucleobase A. It occurs in mRNA, rRNA and snRNAs. So far it has not been studied in the spliceosomal U6 snRNA, where it occurs on the central A of the essential U6-ACAGA box element, which pairs with the pre-mRNA 5' splice site. In the work described in their manuscript, the authors exploit the fact that U6 is the sole target for m6A modification in *S. pombe*. They use a strain in which the modification enzyme mtl16 is deleted, and they perform exhaustive RNA-seq to investigate any effects on splicing that may be caused by a lack of m6A in U6. They report that the splicing of introns containing an A at the fourth position was affected in the mutant strain. This intron position pairs with the central A of the U6-ACAGA box bearing the m6A under wild-type conditions. Corroborating this idea, introns with a U at this position were not affected in the mutant strain, suggesting that a U-A pair is more stable than an A-A pair in the context of the mutant. The exon upstream of the 5' splice site is paired with the U5 snRNA 5' stem loop. Therefore, the authors searched for possible stabilizing effects from this pairing in the destabilized A-A mutants. Using mutant U5 snRNAs they indeed found that increasing exon/U5 pairing decreases missplicing in selected model constructs.

The paper presents novel insight into the m6A modification of U6 snRNA. The experiments, though complex are well presented in a logical way, allowing clear understanding of the arguments. However, one technical point and two conceptual aspects, should be addressed in a revision.

The technical point relates to the thresholds used in a number of figure panels (orange lines) that delineate the true deviants in the analysis. First, the Materials and Methods section (pp. 17–18) outlines the procedure used, but apparently only that for Figure 2C. However, this reads much like adjusting parameters to obtain the desired results. This should be rather driven by a clear statement of the criteria for, say, a misspliced intron, followed by extraction of a high-confidence set of misspliced introns. I was particularly puzzled as to where the numbers in the inequality (p. 18, line 11) come from. In further figures, the definition of the orange threshold lines is outlined in the legends and is "redefined" each time. This point needs clarification, so that the calculations can be followed.

My second point concerns the interpretation. While it makes perfect sense to explain all effects in terms of base-pairing between the U6-ACAGA box and the 5' end of the intron, the non-canonical nature of the A-A pair is not touched upon. Thermodynamic data can certainly be derived from model substrates (as discussed on pp. 10–11), but the authors do not justify their application in the present case. It can also be envisaged that the m6A modification interferes with different aspects of the splicing reaction, such as an essential structural transition. For example, in the B complex (human: 5o9z) it looks as though there is a base-pairing interaction, whereas in the ILS complex (*S. pombe*: 3jb9) it looks as though A37 has shifted out of this pairing to become located between G103 and U104, the latter being the "A4" position. Alternative interpretations are worth discussing, even if they are not favored. They would certainly enrich the manuscript.

My third point concerns the highly compressed narrative in the introduction. In the last paragraph there is confusion as to what organism is being discussed. From a description of the m6A in U6 in human and *S. pombe*, mouse data are added without the mouse being mentioned at all. Also, the most important aspect of the paper, namely base-pairing of the U6-ACAGA box to the 5' end of the intron is not explained in detail at this stage. To address this, the manuscript is in need of focused restructuring.

Minor points:

Fig 1a: What is the underlined sequence?

Fig 1b and S4: Explain "n.d.".

p4, bottom: What does the slower growth on the nonfermentable carbon source imply?

p8, 2nd. paragraph: Table S1 should be Supplementary Table S1

p10, end of 1st. paragraph: How is the spliceosome supposed to "monitor" the 5' splice-site interaction with U5 and U6 snRNAs? I think it is a long jump to the next sentence about evolution. Furthermore, I doubt whether consideration of four organisms (Fig. S12) warrants evolutionary conclusions.

p19, top paragraph: The "gel" is already in the abbreviation PAGE.

p20: What is "Cp"?

Fig 3d: The inset should be placed separately.

Fig 5a: Why does the threshold not start at the origin?

Fig S7B: Replicates should be indicated on the axis.

Fig S7A and B: Why are there no "WT vs. mutant" plots?

Reviewer #3 (Remarks to the Author):

This manuscript presents an analysis of splicing events in a pombe strain lacking *mtl16*, an RNA methyltransferase that installs m6A on U6 snRNA. U6 is involved in recognition of the splice site during the dynamics of the spliceosome. Its m6A37 interacts with the 5'SS, downstream of the exon-intron junction thus providing a plausible rationale for altered splice site selection. Like the human homologue METTL16 characterized by the same group, the pombe enzyme can act as stand-alone enzyme on U6 snRNA. Loss of m6A in U6 RNA did not alter its steady state level in vivo.

A comparison of polyA seq data from wt and ko strains indicated differences linked to the absence of the m6A, mostly manifested as intron retention. The observed differences could be segregated according to frequency, into groups characterized by a consensus sequence in the part of the intron interacting with the methylated part of U6. Of particular interest where sequences containing an A at position 4 interaction with m6A, An adjacent trinucleotide AAG sequence was investigated in mutational control experiments in minigenes, which suggested that this triplet might mediated a rescue effect via the U5 snRNA.

The resulting model implies a very interesting function of the modified m6A residue, which is mediated by a structural effect of the methylgroup. The model and thus the topic are intriguing and timely, but not validated to a great extent. Especially the interesting component of the structural effect of m6A37 with the opposing A is mostly speculation.

I think this is an interesting story in principle, but a structural biology component characterizing said interaction should be added before publication.

First of all, we appreciate all reviewers for spending your precious time to review our manuscript and giving us a number of productive comments to improve it.

Here are our point-by-point replies to the concerns raised by the referees. Revised descriptions in the main text are marked in yellow.

Response to Reviewer #1's comments

In this manuscript, Ishigami and co-workers examine the role of an m6A modification on the U6 snRNA and its impact on splicing efficiency. The authors begin with a demonstration that a *mtl16* knockout strain lacks the m6A mark on the U6 snRNA by Mass Spectrometry. They further demonstrate that cells lacking *mtl16* are more susceptible to cellular stress but that it is not due to a change in amount of snRNPs present in the cell. To better understand the role of the m6A modification of U6 on splicing, they perform RNAseq to identify splicing events that are affected in the absence of *mtl16*. Authors' analysis of these data suggest to them a difference in the behavior of introns containing a T at position 4 of the intron relative to those containing an A. Authors then build a model correlating sequences at the 3' end of the upstream exon and the nucleotide identity at position 4 of the intron and how methylation of U6 would impact these different events.

We appreciate deep understanding of our works in this study.

Although I find authors' hypotheses about the role of this m6A in splicing to be quite intriguing, I cannot recommend publication of this work in its current state because of the absence of any statistical rigor in the analysis of the data presented here, and the errors that likely derive from this absence. It is well understood today that the 'noise' that is present in a quantitative sequencing experiment is not equally present for all measured individuals in an experiment, but rather is correlated with the level of detection of the individual: highly sampled events are subject to lower variance than are poorly sampled events. This phenomenon is well described in Anders and Huber, *Genome Biology* 2010. In the current work, authors use a 'home-made' method for attempting to assess meaningful changes, wherein they assess (among a few other criteria) whether " $PSE\ KO - PSE\ WT > 2 \times SD$ " (page 17, I also note that authors have an error in their formula for SD on this page in that they are missing the $\wedge 2$ for both of the individual variance values). There are at least two important problems with this approach. First, for an experiment such as this where there are only two replicates, the

use of 2 SD units is massively below the requirement for using a standard t-test to assess significance. Very briefly, this is the reason that analysis tools such as dEseq use estimates of global (expression-dependent) variance in their testing rather than measured variance for individuals in their approach.

Second, and equally importantly, the approach authors take incorrectly assumes that variance (SD) will be consistent across the spectrum of samples. But because variance is correlated with overall sampling level, lowly expressed samples will naturally be subject to greater variance, making it more difficult to ‘significantly’ detect a defect in splicing according to these criteria. In the plot below, I have taken the raw data (author provided) and calculated the coefficient of variation (SD/Mean) of the EIJ metric (for the CON1 and CON2 measurements) for each (non-zero) event in the ‘CON’ dataset and plotted these values as a function of the ‘EIJCON1’ value.

(see attached file)

As expected, there is significantly higher variance associated with the lower sampled individuals (this is particularly apparent as count numbers decrease from 100 \rightarrow 1, there is also a surprising anomaly in the data above \sim 100 which suggests an artifact in normalization and or compression). The problem with the approach that authors have used here is that it is much more difficult according to their formula to detect a lowly sampled event (those with higher overall SD) than for a highly expressed event (with lower overall SD). This problem is particularly important given that authors’ data also show that global expression values for introns containing a T at position 4 are higher than are global expression values for introns containing an A at position 4, the combined result being that authors’ statistical approach makes it more likely to identify T4 introns than A4 introns. This statistical problem makes it impossible to determine whether the result that authors have described in their work is actually a function of the loss of m6A, or simply reflects the increased ability of the T4 introns to pass authors’ statistical filters: but that both T4 and A4 introns are equally impacted by the absence of m6A, contrary to their presented conclusion.

We really appreciate these critical comments on our statistical analyses; we agree that there were many arbitrary threshold settings with very low requirements due to the large variance in the read counts that were shown in your comment, which lead to the indication that the difference in PSE distribution is not caused by the sequence characteristics of the introns but by the larger variance in introns with smaller mapped number of reads.

We assume that the variance in read counts were large because the harvest

timing of the two replicates were a year apart. The large variation between two replicates is probably due to the lot difference of culture media (probably yeast extract). To solve this issue, we prepared new sets of biological replicates to replace the data of the previous version of the manuscript, with an increased number of replicates from 2 to 4. This gave a large decrease in the CV, as shown in the figure below.

The plot among EIJCON1 count of the first quadruplicate against the CV of EIJCON was redrawn using the new data, which showed a much smaller CV, especially in the read count range between 10 and 100. As for the statistic problem that the requirement is massively below the standard, we declared a rigor threshold which applies Welch's t-test against differences in PSE with its p-value adjusted for multiple tests by Storey's Q-value method. Normality of PSE was verified by Shapiro-Wilk test and Quantile-Quantile plot. The conclusion was not changed by these new experiments and analyses.

Response to Reviewer #2's comments

One widespread RNA modification is N6-methylation of the nucleobase A. It occurs in mRNA, rRNA and snRNAs. So far it has not been studied in the spliceosomal U6 snRNA, where it occurs on the central A of the essential U6-ACAGA box element, which pairs with the pre-mRNA 5' splice site. In the work described in their manuscript, the authors exploit the fact that U6 is the sole target for m6A modification in *S. pombe*. They use a strain in which the modification enzyme *mtl16* is deleted, and they perform exhaustive RNA-seq to investigate any effects on splicing that may be caused by a lack of m6A in U6. They report that the splicing of introns containing an A at the fourth position was affected in the mutant strain. This intron position pairs with the central A

of the U6-ACAGA box bearing the m6A under wild-type conditions. Corroborating this idea, introns with a U at this position were not affected in the mutant strain, suggesting that a U-A pair is more stable than an A-A pair in the context of the mutant. The exon upstream of the 5' splice site is paired with the U5 snRNA 5' stem loop. Therefore, the authors searched for possible stabilizing effects from this pairing in the destabilized A-A mutants. Using mutant U5 snRNAs they indeed found that increasing exon/U5 pairing decreases missplicing in selected model constructs.

The paper presents novel insight into the m6A modification of U6 snRNA. The experiments, though complex are well presented in a logical way, allowing clear understanding of the arguments.

We appreciate deep understanding of our work and positive comments.

However, one technical point and two conceptual aspects, should be addressed in a revision.

The technical point relates to the thresholds used in a number of figure panels (orange lines) that delineate the true deviants in the analysis. First, the Materials and Methods section (pp. 17–18) outlines the procedure used, but apparently only that for Figure 2C. However, this reads much like adjusting parameters to obtain the desired results. This should be rather driven by a clear statement of the criteria for, say, a misspliced intron, followed by extraction of a high-confidence set of misspliced introns. I was particularly puzzled as to where the numbers in the inequality (p. 18, line 11) come from. In further figures, the definition of the orange threshold lines is outlined in the legends and is “redefined” each time. This point needs clarification, so that the calculations can be followed.

We agree that the multiple rounds of threshold setting are confusing. As pointed out by Reviewer #1, we realized huge variation between duplicate data sets of RNA-seq in the original manuscript. To solve this problem, we re-obtained quadruplicate data sets of RNA-seq using freshly prepared RNA samples from WT and KO strains, and replaced all the data to ensure statistical reliability in this revised manuscript. As mentioned above, the new data set gives a CV value much smaller than the previous data set. With this new dataset, the statistical analyses have been improved so that the thresholds are defined only once, with an arbitrary value of 10 for difference in PSE and 5 for difference in PA5S and PA3S between WT and KO strains.

My second point concerns the interpretation. While it makes perfect sense to explain all effects in terms of base-pairing between the U6-ACAGA box and the 5' end of the intron, the non-canonical nature of the A-A pair is not touched upon. Thermodynamic data can certainly be derived from model substrates (as discussed on pp. 10–11), but the authors do not justify their application in the present case.

We recognized that the thermodynamic data only derived from model substrates with sequences different from the spliceosome context. In this revision, we measured melting temperature of short hairpins that mimic base-pairing between the U6-ACAGA box and the 5' end of the intron (Figure 3c). As shown in Figure 3d, m⁶A at position 37 in the middle of ACAGA box slightly but significantly stabilizes the duplex bearing A at position 4 of the intron by increasing T_m value with 0.34°C, but destabilizes the duplex bearing U at the same site by decreasing T_m value with 0.66°C. We here obtained a biochemical evidence to prove that m⁶A actually stabilizes non-canonical A-A pair in the context of U6-5'SS pairing.

It can also be envisaged that the m⁶A modification interferes with different aspects of the splicing reaction, such as an essential structural transition. For example, in the B complex (human: 5o9z) it looks as though there is a base-pairing interaction, whereas in the ILS complex (S. pombe: 3jb9) it looks as though A37 has shifted out of this pairing to become located between G103 and U104, the latter being the “A4” position. Alternative interpretations are worth discussing, even if they are not favored. They would certainly enrich the manuscript.

We appreciate this suggestion. The intron retention is the major problem in mRNA splicing observed in the *mll16*Δ strain, indicating that m⁶A in U6 snRNA plays a role in the first transesterification in the splicing reaction. In fact, we observed different tendency of splicing defect between A4 and U4 introns. The thermodynamics experiment also demonstrated that m⁶A slightly but significantly stabilizes the duplex with A4 intron. If m⁶A does not base pair with A4 in the 5'SS in the B complex, m⁶A should stabilize the duplex of U6-ACAGA box and the intron through base stacking and not by a putative m⁶A-A pairing. As pointed out by this reviewer, the m⁶A seems to take a different position at the ILS complex; we suggest that this is because the spliceosome went numerous alterations in conformation from B complex to ILS complex, and the structure around the U6 snRNA-5'SS duplex might have been altered in these steps.

This interpretation was added to the first and second paragraphs in the discussion.

My third point concerns the highly compressed narrative in the introduction. In the last paragraph there is confusion as to what organism is being discussed. From a description of the m6A in U6 in human and *S. pombe*, mouse data are added without the mouse being mentioned at all. Also, the most important aspect of the paper, namely base-pairing of the U6-ACAGA box to the 5' end of the intron is not explained in detail at this stage. To address this, the manuscript is in need of focused restructuring.

The 5th paragraph of the introduction was restructured so that it is clear which organism we are discussing about. We here discuss function of mouse Mett16 in the 4th paragraph.

Minor points:

Fig 1a: What is the underlined sequence?

This is the sequence that base pairs with 5' end of intron. This information was added to the legend.

Fig 1b and S4: Explain "n.d."

"n.d." stands for non detected. It was added in the legend.

p4, bottom: What does the slower growth on the nonfermentable carbon source imply?

It implies deficiency in respiration caused by disorders in mitochondria; the explanation was added to the manuscript.

p8, 2nd. paragraph: Table S1 should be Supplementary Table S1

It was fixed.

p10, end of 1st. paragraph: How is the spliceosome supposed to "monitor" the 5' splice-site interaction with U5 and U6 snRNAs? I think it is a long jump to the next sentence about evolution.

The cryoEM structures of spliceosomes indicate that the interaction between pre-mRNA and U5/U6 snRNAs juxtaposes the 5'SS to the catalytic active site in the B complexes. Then, the Prp2 (*S. pombe* Cdc28) helicase pulls the intron to bring the branch site to the active site to form the B* complex. The process requires large structural rearrange the spliceosome. During this process, the 5'SS needs to be anchored firmly to the active site, enabling the branch site 2'-OH to attack the splice site of 5'SS. We speculate that stable recognition of 5'SS by U5 and U6 snRNAs decides the overall efficiency of the first transesterification. This discussion was applied to the 1st paragraph of the discussion. The sentence structure was changed to eliminate the long jump.

Furthermore, I doubt whether consideration of four organisms (Fig. S12) warrants evolutionary conclusions.

We added several other organisms to Supplementary Figure 15 (originally S12).

p19, top paragraph: The “gel” is already in the abbreviation PAGE.

It was fixed.

p20: What is “Cp”?

It is an abbreviation of “crossing point” used in the Roche software. An explanation was added to the manuscript.

Fig 3d: The inset should be placed separately.

The boxplot was placed outside of the cumulative curve plot.

Fig 5a: Why does the threshold not start at the origin?

Fig S7B: Replicates should be indicated on the axis.

Fig S7A and B: Why are there no “WT vs. mutant” plots?

All the data of RNA-seq were replaced with new ones in this revision. Thus, all these

figures are replaced, and their statistical analyses are also improved.

Response to Reviewer #3's comments

This manuscript presents an analysis of splicing events in a pombe strain lacking *mtl16*, an RNA methyltransferase that installs m⁶A on U6 snRNA. U6 is involved in recognition of the splice site during the dynamics of the spliceosome. Its m⁶A³⁷ interacts with the 5'SS, downstream of the exon-intron junction thus providing a plausible rationale for altered splice site selection.

Like the human homologue METTL16 characterized by the same group, the pombe enzyme can act as stand-alone enzyme on U6 snRNA. Loss of m⁶A in U6 RNA did not alter its steady state level in vivo.

A comparison of polyA seq data from wt and ko strains indicated differences linked to the absence of the m⁶A, mostly manifested as intron retention.

The observed differences could be segregated according to frequency, into groups characterized by a consensus sequence in the part of the intron interacting with the methylated part of U6. Of particular interest were sequences containing an A at position 4 interaction with m⁶A. An adjacent trinucleotide AAG sequence was investigated in mutational control experiments in minigenes, which suggested that this triplet might mediate a rescue effect via the U5 snRNA.

The resulting model implies a very interesting function of the modified m⁶A residue, which is mediated by a structural effect of the methyl group. The model and thus the topic are intriguing and timely, but not validated to a great extent. Especially the interesting component of the structural effect of m⁶A³⁷ with the opposing A is mostly speculation.

I think this is an interesting story in principle, but a structural biology component characterizing said interaction should be added before publication.

We appreciate deep understanding of our work and positive comments. As you pointed out, the original manuscript lacked structural aspect of m⁶A-A interaction in the U6 snRNA-5'SS duplex. In this revision, we measured melting temperature of short hairpins that mimic base-pairing between the U6-ACAGA box and the 5' SS of the

intron (Figure 3c). As shown in Figure 3d, m⁶A slightly but significantly stabilizes the duplex with A4 intron by increasing T_m value. We here obtained a biochemical and thermodynamic evidence to prove that m⁶A actually stabilizes non-canonical A-A pair in the context of U6 snRNA-5'SS pairing. This experiment should support the raised speculation of the function of m⁶A.

Reviewers' comments:

Reviewer #1 (Remarks to the Author):

Apologies for my delay in returning this review. I appreciate authors' extensive efforts at recollecting and reanalyzing their experiments. To be sure, the quality of the data included here are significantly improved from the earlier version. As during my initial review, I continue to find authors' hypothesis regarding the role of m6A in U6 to be quite interesting. Importantly, however, as described below there remains a critical statistical error in the analysis of this work which should preclude its publication in its current state. Nevertheless, I do believe the underlying data support at least some of the conclusions presented here, and I describe below an appropriate analytical pathway which should allow authors to revise their analysis in a way that may provide important insights for the field. There are two critical defects in the analytical approach used by authors which have important consequences for proper understanding of the data. The first regards the first part of authors' decision as described on pg. 7 to classify as splicing defective those "introns with a PSE difference larger than 10 and a significant difference between the WT and mtl16Δ strains". While I realize that this approach is used by other groups within the field, it is nevertheless a statistically unsound method by which to analyze the data because of the bias that it introduces. By requiring a difference in PSE of greater than 10 (mathematically: $|PSE_{WT} - PSE_{mtl16\Delta}| > 10$), authors have created a criterion which some events can satisfy more easily than others, while others still cannot satisfy it at all. For example, for introns with a PSE_{WT} of 3.5 (the ~median value in these data) it is impossible for PSE_{mtl16Δ} to ever decrease in value by 10 because all PSE values are naturally restricted between 0 and 100 [[Here I note that equation (2) on page 6 is incorrectly written relative to how it is described in the text and the supplementary data, and should instead read: $PSE = ((cov(Total) - cov(CSR)) / (cov(Total)) * 100)$]. More importantly, an intron with a PSE_{WT} of 1 (about the 10th percentile of the data) would need to increase to 11 to be considered significant, a change of 11-fold in apparent splicing efficiency; by contrast, an intron with a PSE_{WT} of 50 would only need to increase to 60 to be considered significant, a change of only ~20% in splicing efficiency. The result of this is that it is much easier to identify 'significant' defects in introns that are naturally poorly spliced. Indeed, this bias can be readily seen within authors' data (from Supplemental Table 1), as demonstrated in the figure below. Panel A is a recreation of Figure 2b in authors' manuscript. Panel B includes the same data, but showing only data from the WT strain: here it (incorrectly) appears that there is no bias in the likelihood of an intron being deemed affected as a function of its 'natural' splicing efficiency (PSE); that is, the red points appear to be evenly distributed about the y-axis. However, Panel C shows these same data where the y-axis is presented in log-scale. Two important points become apparent here: first is that the overall behavior of PSE values approximates a normal distribution when viewed in logarithmic space (although the hard limit imposed at both the low (0) and high (100) ends importantly means that they are not normally distributed: see below for more), and second is that there is a strong bias in the splicing efficiency of the introns authors deemed affected in the mutant strain. Indeed, whereas 50% of all introns have a 'natural' PSE of less than 3.5%, 90% of all 'significant' events identified by authors have a 'natural' PSE of greater than 3.5%. The impact of this bias is really important in terms of the biological conclusions, and best illustrated when considering individual splicing events. Panel D shows the raw counts for intron3 of ckb1, an intron with an A at position 4 and which was deemed 'significant' by authors' analysis. By contrast, panel E shows the raw counts for intron3 of nup45, an intron with a T at position 4 but which was NOT deemed significant by authors' analysis. IMPORTANTLY, whereas authors' analysis suggested that between these two, only the ckb1 intron (an i4A variant) was mis-spliced, there is in fact stronger statistical evidence supporting the mis-splicing of the nup45 intron (an i4T variant). The broader concern then, as it was upon initial submission of this work, is whether the conclusion that loss of the m6A in U6 is mostly impacting i4A introns is biologically meaningful or simply an artifact of this problem in data analysis. As described later, I think there is good reason to believe that the underlying hypothesis is correct, but supporting this conclusion requires the use of a robust statistical approach that will faithfully identify all of the mis-regulated events in this mutant background, regardless of their 'natural' splicing efficiency, coupled with a subsequent analysis of those affected events.

The second major concern regards the second part of authors' decision as described on pg. 7 to classify as splicing defective those "introns with a PSE difference larger than 10 and a significant difference between the WT and mtl16Δ strains". Although the authors claim that "PSE values follow normal distribution", their empirical support for this (shown in Supp. Figure 8) examines only a small fraction of the data. Because of the mathematical construction of PSE [[taking the general form of: $(x-y)/x$, where $0 \leq y \leq x$]], wherein values of PSE are constrained between 0 and 1 (or 100 as noted above), PSE values cannot conform to the Normal distribution, which by definition is homoscedastic and requires infinitely long 'tails' to the left and right. The figure to the right shows the QQ plot for the full set of PSE_{WT} data, which highlights both the extent to which these data are non-Normal, and the lack of 'significant' introns at the edges of the PSE range. Authors are on the right track in Supplement Fig. 8 in using a log-transformation of the data, as well in using Welch's t-test (also see later); however, (and although I am not a trained statistician) I do not think it is appropriate to apply such a test to a sub-range of a non-Normal dataset, even if that subrange appears Normally distributed. Regardless, this work requires a statistical approach that can evaluate events across the entire set of introns, and because neither PSE nor log(PSE) are Normally distributed, this requires something different.

Here I suggest an alternate approach for analyzing this work, the results of which provide compelling support for the larger hypothesis presented herein, albeit with important differences and requirements for additional analysis of the later portions of the manuscript.

An initial suggestion is driven partly by statistical considerations and partly by biological considerations. On page 6, authors present four different equations that they use to describe changes in splicing that might occur within their dataset. As noted above, the construction of these equations is important in terms of how they can be used for statistical testing. From a biological perspective, it is my opinion that the most interesting and important biological conclusion from the work here is that there is a splicing defect in the \neg mtl16Δ \neg strain that leads to a significant level of intron retention. As such, I would suggest that authors largely restrict their analysis of this work to this metric, through an alternative equation for PIR. Regardless, the descriptions here for changes in approach could be equally applied to PSE, PA5S or PA3S. As noted above for PSE, the problem with PIR as described in equation (3) is that it is naturally limited to the range of 0 to 1 (or 100). The origin of this problem is the use of the term cov(Total) in the denominator: because the components of the numerator ((cov(EIJR)+cov(IEJR))/2 for PIR and cov(Total)-cov(CSR) for PSE) are also a part of the denominator, the ranges will always be limited. Importantly, however, because the majority of the reads corresponding for any intron are spliced reads, cov(CSR) becomes the major component of cov(Total). With the simple change of calculating PIR as $(\text{cov}(\text{EIJR}) + \text{cov}(\text{IEJR})) / (2\text{cov}(\text{CSR}))$, both the numerator and the denominator can independently range between 0 and ∞ . As such, using $\text{PIR} = \log_2 ((\text{cov}(\text{EIJR}) + \text{cov}(\text{IEJR})) / (2\text{cov}(\text{CSR})))$ should provide a Normally distributed set of values representing the relationship between spliced and unspliced isoforms across the entire genome. Indeed, the figure to the right shows the mean of PIR_{WT} values calculated this way from authors' dataset, and the confirmation of its Normal distribution. With the data in this format, it is statistically plausible to use a test such as Welch's t-test to identify introns that are statistically significantly different between the WT and mtl16Δ strains.

Using this approach, one can compare the PIR_{WT} and PIR_{mtl16Δ} strains using Welch's t-test, the results of which are shown in the figure below. It is in looking at the data in this way that a different biological conclusion becomes apparent than what is presented by authors. Panel A shows the difference between the PIR_{WT} and PIR_{mtl16Δ} strains, and here a strong argument could be made that nearly every splicing event is negatively impacted in the mtl16Δ strain. There is of course the important question of statistical significance of any individual result, but this is in fact a quite striking view of the data, and there can be little argument that the overall behavior of the mtl16Δ strain is that there is a global increase in the levels of unspliced mRNA. This is in fact contrary to authors' claims that "whereas very few T4 introns ... were affected by the m6A loss." An important question then is which of these changes is statistically significant? To consider significance, Panel B shows these data where the significance is plotted on the y-axis (calculated with Welch's t, and shown as $-\log(p)$). The line corresponding to $p=0.05$ is shown in the plot (without any multiple hypothesis correction yet). A major concern in analyzing an RNAseq experiment such as this is that many splicing events are

sampled too poorly (low counts) to be able to generate a statistically significant result (too much variance in lowly sampled species means that their t-scores are never high enough). Indeed, when comparing those events meeting the significance threshold (above the line) with those below it, the major difference is not in the fold change, but in the variance associated with the measurement. As such, it is likely that a (much) larger sized sequencing experiment would in fact find that most of the introns would be detected as significantly defective. I admit here that my expectation when reading the original version of this manuscript (and the current version) was that there would be a difference in expression level between i4A and i4T introns, such that the i4A introns are more highly detected and therefore are more easily found above the significance threshold: that the conclusion drawn by authors was an artifact of the sequencing depth included in the study (and as well the analytical method). But importantly, Panel C shows the mean counts of EIJR and IEJR in the WT sample, and this shows that in fact i4T introns are not significantly lower expressed on average than i4A introns. By contrast, Panel D shows that the fold change observed for i4A is globally stronger than what is seen for i4T introns. To summarize then, I think that authors' data provide a compelling argument that the loss of m6A in U6 causes a defect in the splicing of ~ALL introns, but the impact is felt most strongly by those with an A at position 4.

An important and difficult question then regards how to address concerns of multiple hypothesis testing, with the goal of assessing enrichment of sequences, etc. Most approaches that have been developed for such a problem are not well suited to considering data such as these, because most of the measured events here are not centered about zero. As such, I would suggest an alternative approach. Rather than trying to find an (arguably) arbitrary cutoff between significant and non-significant, I would suggest separating the data into quantiles (where n might be 4 or 6 or 10) according to the confidence in their deviation. The easiest way to quantify this confidence would be with some sort of a Z-score or t-score: for each event, consider the difference in mean divided by the standard deviation (or SEM). When considered this way, for example, the top quartile of Z-scores is composed of 932 i4A introns (out of 1285 total in the quartile): using Fisher's Exact, this enrichment is significant at <0.00001 . Analyzing the data in this way would allow authors to find natural differences in properties without requiring a hard cutoff. Importantly, analyzing the data this way while using an appropriate statistic framework seems likely to support the larger hypothesis about the interaction of the m6A with i4A introns, but the 'rest' of the paper would need to be resolved with a re-analysis of the data generated by this approach.

As one final note then, I return to the idea of focusing analysis on PIR. Although authors have included an analysis of PA5S in the current manuscript, I would argue that this artificially creates a difference in terminology when there is nothing novel about the mechanism. The concept of 'alternative' or 'cryptic' splice sites are entirely a function of 'our' desire to annotate biology. But from the perspective of the larger organism, it is simply the case that at some frequency the spliceosome performs chemistry at a location within an mRNA that corresponds to what we consider the 'right' location, and at a different frequency it activates a different site. With this view, authors' observation that the *mtl16Δ* strain shows increased PA5S is not mechanistically any different from what we have been discussing as PIR for otherwise canonical introns. Indeed, it would be entirely expected that a defect in the splicing machinery that impact some aspect of 5'SS recognition would similarly impact both canonical and 'cryptic' splice sites. As noted earlier, a consequence of this is that I think there is a really interesting biological story here that would greatly benefit from focusing on the impact of this important mutant on global PIR.

Reviewer #2 (Remarks to the Author):

Following the criticism of reviewer #1, the authors redid the experiments in independent quadruplicates. The outcome after recalculating the data is that they lose roughly one-half of their original events related to splicing errors (PSE). This is apparent in the comparison of Fig 2b (6.90 % corresponds 355 events; the legend (p 25, ll 804-5) probably should read 'dot/s' for 'plot/s') of the revised manuscript to Fig 2c (16.94 % corresponds to 782 events) of the original submission and

threads through all subsequent analysis. At face value, this implies that the data generated are not robust and may vary more than 2-fold from experiment to experiment. Comparison of Figs 3a of the revised and original manuscript would suggest that ca. 373 A4 introns are no longer responsive to ablation of *mtl16*. This is worrisome, in particular as the authors attribute the problems in the first data set to the use of different batches of yeast extracts. That may be the case, but it does not help in understanding the N6-methylation of the spliceosomal U6 snRNA at position A37 in *S. pombe*.

There is an issue with documenting the statistical procedures. While the authors say that they 'developed an algorithm to detect non-canonical splicing events' (p6, revised and original), the algorithm as such is not explained. This should be elaborated in detail in a supplementary section, taking care to explain how the calculations were performed. Were they done with R, a spreadsheet calculation, or some other tool? Also, it is not clear whether the algorithm is publicly available on dedicated platforms.

The thermodynamic data on the artificial RNA constructs (P12, ll 392-5, Fig 3 c and d) are in agreement with expectations, but I think it is an overstatement to say that the 'thermodynamic data were reproduced in the sequence context of the spliceosome'.

The revision itself appears to be a preliminary draft: the references are completely missing, the in-text references are not formatted, and some are not yet put in place (p21, l 678). The authors should exercise more care when submitting a revision.

Reviewers' comments:

Reviewer #1 (Remarks to the Author):

Apologies for my delay in returning this review. I appreciate authors' extensive efforts at recollecting and reanalyzing their experiments. To be sure, the quality of the data included here are significantly improved from the earlier version. As during my initial review, I continue to find authors' hypothesis regarding the role of m6A in U6 to be quite interesting. Importantly, however, as described below there remains a critical statistical error in the analysis of this work which should preclude its publication in its current state. Nevertheless, I do believe the underlying data support at least some of the conclusions presented here, and I describe below an appropriate analytical pathway which should allow authors to revise their analysis in a way that may provide important insights for the field.

We really appreciate Reviewer #1 for taking time to carry out extensive analysis for the suggestion of a statistical method more suitable for this study. According to this suggestion, we have learned how to analyze our dataset properly with statistical rigor.

There are two critical defects in the analytical approach used by authors which have important consequences for proper understanding of the data. The first regards the first part of authors' decision as described on pg. 7 to classify as splicing defective those "introns with a PSE difference larger than 10 and a significant difference between the WT and *mtl16Δ* strains". While I realize that this approach is used by other groups within the field, it is nevertheless a statistically unsound method by which to analyze the data because of the bias that it introduces. By requiring a difference in PSE of greater than 10 (mathematically: $|PSE_{WT} - PSE_{mtl16\Delta}| > 10$), authors have created a criterion which some events can satisfy more easily than others, while others still cannot satisfy it at all. For example, for introns with a PSE_{WT} of 3.5 (the ~median value in these data) it is impossible for PSE_{mtl16Δ} to ever decrease in value by 10 because all PSE values are naturally restricted between 0 and 100 [[Here I note that equation (2) on page 6 is incorrectly written relative to how it is described in the text and the supplementary data, and should instead read: $PSE = ((cov(Total) - cov(CSR)) / (cov(Total))) * 100$]]. More importantly, an intron with a PSE_{WT} of 1 (about the 10th percentile of the data) would need to increase to 11 to be considered significant, a change of 11-fold in apparent splicing efficiency; by contrast, an intron

with a PSEWT of 50 would only need to increase to 60 to be considered significant, a change of only ~20% in splicing efficiency. The result of this is that it is much easier to identify ‘significant’ defects in introns that are naturally poorly spliced. Indeed, this bias can be readily seen within authors’ data (from Supplemental Table 1), as demonstrated in the figure below. Panel A is a recreation of Figure 2b in authors’ manuscript. Panel B includes the same data, but showing only data from the WT strain: here it (incorrectly) appears that there is no bias in the likelihood of an intron being deemed affected as a function of its ‘natural’ splicing efficiency (PSE); that is, the red points appear to be evenly distributed about the y-axis. However, Panel C shows these same data where the y-axis is presented in log-scale. Two important points become apparent here: first is that the overall behavior of PSE values approximates a normal distribution when viewed in logarithmic space (although the hard limit imposed at both the low (0) and high (100) ends importantly means that they are not normally distributed: see below for more), and second is that there is a strong bias in the splicing efficiency of the introns authors deemed affected in the mutant strain. Indeed, whereas 50% of all introns have a ‘natural’ PSE of less than 3.5%, 90% of all ‘significant’ events identified by authors have a ‘natural’ PSE of greater than 3.5%. The impact of this bias is really important in terms of the biological conclusions, and best illustrated when considering individual splicing events. Panel D shows the raw counts for intron3 of *ckb1*, an intron with an A at position 4 and which was deemed ‘significant’ by authors’ analysis. By contrast, panel E shows the raw counts for intron3 of *nup45*, an intron with a T at position 4 but which was NOT deemed significant by authors’ analysis. IMPORTANTLY, whereas authors’ analysis suggested that between these two, only the *ckb1* intron (an i4A variant) was mis-spliced, there is in fact stronger statistical evidence supporting the mis-splicing of the *nup45* intron (an i4T variant). The broader concern then, as it was upon initial submission of this work, is whether the conclusion that loss of the m6A in U6 is mostly impacting i4A introns is biologically meaningful or simply an artifact of this problem in data analysis. As described later, I think there is good reason to believe that the underlying hypothesis is correct, but supporting this conclusion requires the use of a robust statistical approach that will faithfully identify all of the mis-regulated events in this mutant background, regardless of their ‘natural’ splicing efficiency, coupled with a subsequent analysis of those affected events.

As written below, we completely renewed the analysis method so that the sequence motifs are not biased as pointed out by the reviewer.

The second major concern regards the second part of authors' decision as described on pg. 7 to classify as splicing defective those "introns with a PSE difference larger than 10 and a significant difference between the WT and *mtl16Δ* strains". Although the authors claim that "PSE values follow normal distribution", their empirical support for this (shown in Supp. Figure 8) examines only a small fraction of the data. Because of the mathematical construction of PSE [[taking the general form of: $(x-y)/x$, where $0 \leq y \leq x$]], wherein values of PSE are constrained between 0 and 1 (or 100 as noted above), PSE values cannot conform to the Normal distribution, which by definition is homoscedastic and requires infinitely long 'tails' to the left and right. The figure to the right shows the QQ plot for the full set of PSE_{WT} data, which highlights both the extent to which these data are non-Normal, and the lack of 'significant' introns at the edges of the PSE range. Authors are on the right track in Supplement Fig. 8 in using a log-transformation of the data, as well in using Welch's t-test (also see later); however, (and although I am not a trained statistician) I do not think it is appropriate to apply such a test to a sub-range of a non-Normal dataset, even if that subrange appears Normally distributed. Regardless, this work requires a statistical approach that can evaluate events across the entire set of introns, and because neither PSE nor log(PSE) are Normally distributed, this requires something different.

We changed the equations as recommended below so that the values are not restricted.

Here I suggest an alternate approach for analyzing this work, the results of which provide compelling support for the larger hypothesis presented herein, albeit with important differences and requirements for additional analysis of the later portions of the manuscript.

An initial suggestion is driven partly by statistical considerations and partly by biological considerations. On page 6, authors present four different equations that they use to describe changes in splicing that might occur within their dataset. As noted above, the construction of these equations is important in terms of how they can be used for statistical testing. From a biological perspective, it is my opinion that the most interesting and important biological conclusion from the work here is that there is a splicing defect in the *mtl16Δ* strain that leads to a significant level of intron retention. As such, I would suggest that authors largely restrict their analysis of this work to this metric, through an alternative equation for PIR. Regardless, the descriptions here for changes in approach could be equally applied to PSE, PA5S or PA3S. As noted above for PSE, the problem with PIR as described in equation (3) is

that it is naturally limited to the range of 0 to 1 (or 100). The origin of this problem is the use of the term $\text{cov}(\text{Total})$ in the denominator: because the components of the numerator ($(\text{cov}(\text{EIJR})+\text{cov}(\text{IEJR}))/2$ for PIR and $\text{cov}(\text{Total})-\text{cov}(\text{CSR})$ for PSE) are also a part of the denominator, the ranges will always be limited. Importantly, however, because the majority of the reads corresponding for any intron are spliced reads, $\text{cov}(\text{CSR})$ becomes the major component of $\text{cov}(\text{Total})$. With the simple change of calculating PIR as $(\text{cov}(\text{EIJR})+\text{cov}(\text{IEJR}))/2\text{cov}(\text{CSR})$, both the numerator and the denominator can independently range between 0 and ∞ . As such, using $\text{PIR}=\log_2((\text{cov}(\text{EIJR})+\text{cov}(\text{IEJR}))/2\text{cov}(\text{CSR}))$ should provide a Normally distributed set of values representing the relationship between spliced and unspliced isoforms across the entire genome. Indeed, the figure to the right shows the mean of PIR WT values calculated this way from authors' dataset, and the confirmation of its Normal distribution. With the data in this format, it is statistically plausible to use a test such as Welch's t-test to identify introns that are statistically significantly different between the WT and *mtl16Δ* strains.

As pointed out, PSE has a limited range and is not normally distributed, thus it is inappropriate to apply Welch's t-test to this score. To solve this issue, we followed the recommendation of using the equation to obtain the score:

$$\text{IRS} = \log_2 \left(\frac{\text{cov}(\text{EIJR}) + \text{cov}(\text{IEJR}) + 0.1}{2 \times \text{cov}(\text{CSR}) + 0.1} \right)$$

As mentioned above and carried out the rest of the analyses. (The score was renamed IRS [Intron Retention Score] instead of PIR.)

To start with, we wrote a normal quantile plot using our data, showing that the results were consistent with the calculation.

(Left: Reviewer #1's normal quantile plot, right: our normal quantile plot)

Using this approach, one can compare the PIRWT and PIRmtl16Δ strains using Welch’s t-test, the results of which are shown in the figure below. It is in looking at the data in this way that a different biological conclusion becomes apparent than what is presented by authors. Panel A shows the difference between the PIRWT and PIRmtl16Δ strains, and here a strong argument could be made that nearly every splicing event is negatively impacted in the mtl16Δ strain. There is of course the important question of statistical significance of any individual result, but this is in fact a quite striking view of the data, and there can be little argument that the overall behavior of the mtl16Δ strain is that there is a global increase in the levels of unspliced mRNA. This is in fact contrary to authors’ claims that “whereas very few T4 introns ... were affected by the m6A loss.”

We observed the distribution of the difference in this IRS score between WT and KO by drawing a histogram of $IRS_{KO} - IRS_{WT}$. This gave a slightly different result, but the overall interpretation that the loss of m⁶A in U6 causes a defect in the splicing of all introns were consistent with the reviewer’s observation. (Reviewer #1’s figure is based on $IRS_{WT} - IRS_{KO}$, while we calculated by $IRS_{KO} - IRS_{WT}$.)

(Left: Reviewer #1’s histogram of difference in IRS, right: our histogram)

An important question then is which of these changes is statistically significant? To consider significance, Panel B shows these data where the significance is plotted on the y-axis (calculated with Welch’s t, and shown as $-\log(p)$). The line corresponding to $p=0.05$ is shown in the plot (without any multiple hypothesis correction yet). A major concern in analyzing an RNAseq experiment such as this is that many splicing events are sampled too poorly (low counts) to be able to generate a statistically significant result (too much variance in lowly sampled species means that their t-scores are never high enough). Indeed, when comparing those events meeting the significance threshold (above the line) with those below it, the major difference is not in the fold change, but

in the variance associated with the measurement. As such, it is likely that a (much) larger sized sequencing experiment would in fact find that most of the introns would be detected as significantly defective. I admit here that my expectation when reading the original version of this manuscript (and the current version) was that there would be a difference in expression level between i4A and i4T introns, such that the i4A introns are more highly detected and therefore are more easily found above the significance threshold: that the conclusion drawn by authors was an artifact of the sequencing depth included in the study (and as well the analytical method). But importantly, Panel C shows the mean counts of EIJR and IEJR in the WT sample, and this shows that in fact i4T introns are not significantly lower expressed on average than i4A introns. By contrast, Panel D shows that the fold change observed for i4A is globally stronger than what is seen for i4T introns. To summarize then, I think that authors' data provide a compelling argument that the loss of m6A in U6 causes a defect in the splicing of ~ALL introns, but the impact is felt most strongly by those with an A at position 4.

An important and difficult question then regards how to address concerns of multiple hypothesis testing, with the goal of assessing enrichment of sequences, etc. Most approaches that have been developed for such a problem are not well suited to considering data such as these, because most of the measured events here are not centered about zero. As such, I would suggest an alternative approach. Rather than trying to find an (arguably) arbitrary cutoff between significant and non-significant, I would suggest separating the data into quantiles (where n might be 4 or 6 or 10) according to the confidence in their deviation. The easiest way to quantify this confidence would be with some sort of a Z-score or t-score: for each event, consider the difference in mean divided by the standard deviation (or SEM). When considered this way, for example, the top quartile of Z-scores is composed of 932 i4A introns (out of 1285 total in the quartile): using Fisher's Exact, this enrichment is significant at <0.00001 . Analyzing the data in this way would allow authors to find natural differences in properties without requiring a hard cutoff. Importantly, analyzing the data this way while using an appropriate statistic framework seems likely to support the larger hypothesis about the interaction of the m6A with i4A introns, but the 'rest' of the paper would need to be resolved with a re-analysis of the data generated by this approach.

We calculated the Z-scores of the two-sample Z-test for each intron by the formula:

$$Z = \frac{\mu_{KO} - \mu_{WT}}{\sqrt{\frac{\sigma_{KO}{}^2}{4} + \frac{\sigma_{WT}{}^2}{4}}}$$

where $\mu$ is the mean IRS and $\sigma$ is the standard deviation of the IRS among the quadruplicates in each strain. We sorted the Z-score by descending order, splitted them by quartiles and compared the nucleotide composition in each position around the 5' SS between the introns with Z-scores over the first quartile against the introns with z-scores under the third quartile, and carried out Fisher's exact test. When the nucleotide compositions of the 5' SS between introns with top Z-scores and bottom Z-scores were compared by DiffLogo, we obtained a logo similar to the previous version of the manuscript, with the A and T highlighted at the 4$^{th}$ position and the AAG sequence negatively highlighted in the exon side. (This time we used DiffLogo instead of Two Sample Logo)

The enrichment of the nucleotides were significant, as the enrichment of A at the 4$^{th}$ nucleotide was significant with p-value < 2.2e-16 tested by Fisher's exact, and the depletion of A at the -3 position in the exon was significant with p-value < 0.01. This led us to compare the difference in IRS between various nucleotide combinations in the 5' exon, like the previous version. This also gave a similar result, demonstrating that when there are more BBH nucleotides in the 5' exon, the intron becomes more sensitive to loss of m$^6$A.

(The significances of different distribution by Steel–Dwass test are shown on the right. * $p < 0.05$, ** $p < 0.01$, *** $p < 0.0005$.)

Overall, the reviewer's statement that "analyzing the data this way while using an appropriate statistic framework seems likely to support the larger hypothesis about the interaction of the m⁶A with i4A introns" was shown to be true, and we further showed that the interaction with the exon nucleotides were also shown to be consistent with the previous version.

As one final note then, I return to the idea of focusing analysis on PIR. Although authors have included an analysis of PA5S in the current manuscript, I would argue that this artificially creates a difference in terminology when there is nothing novel about the mechanism. The concept of 'alternative' or 'cryptic' splice sites are entirely a function of 'our' desire to annotate biology. But from the perspective of the larger organism, it is simply the case that at some frequency the spliceosome performs chemistry at a location within an mRNA that corresponds to what we consider the 'right' location, and at a different frequency it activates a different site. With this view, authors' observation that the *mtl16Δ* strain shows increased PA5S is not mechanistically any different from what we have been discussing as PIR for otherwise canonical introns. Indeed, it would be entirely expected that a defect in the splicing machinery that impact some aspect of 5'SS recognition would similarly impact both canonical and 'cryptic' splice sites. As noted earlier, a consequence of this is that I think there is a really interesting biological story here that would greatly benefit from focusing on the impact of this important mutant on global PIR.

We deleted the whole section describing altered "alternative" splicing.

We would like to thank the reviewer again for giving us these suggestions on statistical analyses, which lead to obtaining results supporting our large story with an improved method.

Reviewer #2 (Remarks to the Author):

We want to thank Reviewer #2 for handling our manuscript again and giving us important comments.

Following the criticism of reviewer #1, the authors redid the experiments in independent quadruplicates. The outcome after recalculating the data is that they lose roughly one-half of their original events related to splicing errors (PSE). This is apparent in the comparison of Fig 2b (6.90 % corresponds 355 events; the legend (p 25, ll 804-5) probably should read ‘dot/s’ for ‘plot/s’) of the revised manuscript to Fig 2c (16.94 % corresponds to 782 events) of the original submission and threads through all subsequent analysis. At face value, this implies that the data generated are not robust and may vary more than 2-fold from experiment to experiment. Comparison of Figs 3a of the revised and original manuscript would suggest that ca. 373 A4 introns are no longer responsive to ablation of *mtl16*. This is worrisome, in particular as the authors attribute the problems in the first data set to the use of different batches of yeast extracts. That may be the case, but it does not help in understanding the N6-methylation of the spliceosomal U6 snRNA at position A37 in *S. pombe*.

In this comment, only the number of introns picked up were compared between the two analyses, ignoring the different statistical method and threshold among them. The analysis on the revised version was meant to have a stricter threshold, resulting in a smaller number of introns picked up. If the threshold of the first analysis ($PSE_{KO} > PSE_{WT} \times 1.6 + 1.34$ and $PSE_{KO} - PSE_{WT} > 2 \times SD$, minimum $cov(Total) > 50$) was applied to 2 out of 4 replicates from the data in the second version of the manuscript, it would pick up 690/4982 introns, which is a small difference to the initial dataset. Thus, we consider that the difference in the number of introns picked up is due to the difference of the analysis method, and is not an inconsistent differential splicing under different batches of yeast extracts.

There is an issue with documenting the statistical procedures. While the authors say that they ‘developed an algorithm to detect non-canonical splicing events’ (p6, revised and original), the algorithm as such is not explained. This should be elaborated in detail in a supplementary section, taking care to explain how the calculations were performed. Were they done with R, a spreadsheet calculation, or some other tool? Also, it is not clear whether the algorithm is publicly available on dedicated platforms.

We uploaded the python scripts to GitHub and made it open to the public.

The thermodynamic data on the artificial RNA constructs (P12, ll 392-5, Fig 3 c and d) are in agreement with expectations, but I think it is an overstatement to say that the 'thermodynamic data were reproduced in the sequence context of the spliceosome'.

We applied a weaker expression in the sentence.

The revision itself appears to be a preliminary draft: the references are completely missing, the in-text references are not formatted, and some are not yet put in place (p21, l 678). The authors should exercise more care when submitting a revision.

We apologize for not including the reference list in the revised draft, we have checked it more carefully.

REVIEWERS' COMMENTS

Reviewer #1 (Remarks to the Author):

I think the modifications that authors have made to the manuscript make this a very compelling manuscript, and I would support its publication. I have only a few small suggestions for authors to consider.

Whereas I think the analysis now demonstrates the broad impact of the mttl16 variant, there remain a few places where the text might be modified to more carefully reflect the A4 result. In particular, in the Summary (lines 22,23), perhaps "... the most significantly impacted introns were enriched for adenosine at the fourth ..." Likewise, a similar modification could be included in the Introduction section, line 92.

On line 204, authors describe analyzing those introns "with a sufficient read depth", but I wasn't able to find any description of how this level of read depth was determined. Based on the numbers included in the sentence, it isn't clear to me that there are any (many?) missing introns. Some additional clarity here, and/or in the methods section about how this was considered by the authors would be useful.

Jeff Pleiss

We really appreciate Reviewer #1 for taking time to carry out extensive analysis and careful investigation of this manuscript. According to his guidance, we have learned how to analyze our dataset properly with statistical rigor and improved it substantially.

REVIEWERS' COMMENTS

Reviewer #1 (Remarks to the Author):

I think the modifications that authors have made to the manuscript make this a very compelling manuscript, and I would support its publication. I have only a few small suggestions for authors to consider.

Whereas I think the analysis now demonstrates the broad impact of the mtl16 variant, there remain a few places where the text might be modified to more carefully reflect the A4 result. In particular, in the Summary (lines 22,23), perhaps "... the most significantly impacted introns were enriched for adenosine at the fourth ..." Likewise, a similar modification could be included in the Introduction section, line 92.

As suggested, we have rephrased them accordingly.

On line 204, authors describe analyzing those introns "with a sufficient read depth", but I wasn't able to find any description of how this level of read depth was determined. Based on the numbers included in the sentence, it isn't clear to me that there are any (many?) missing introns. Some additional clarity here, and/or in the methods section about how this was considered by the authors would be useful.

It has been clarified with description on the exact number of introns picked up, showing that the number of missing introns is minimum.